# LaSCal: Label-Shift Calibration without target labels

**Teodora Popordanoska**[*]    **Gorjan Radevski**[*]    **Tinne Tuytelaars**    **Matthew B. Blaschko**

ESAT-PSI, KU Leuven
`firstname.lastname@kuleuven.be`

## Abstract

When machine learning systems face dataset shift, model calibration plays a pivotal role in ensuring their reliability. Calibration error (CE) provides insights into the alignment between the predicted confidence scores and the classifier accuracy. While prior works have delved into the implications of dataset shift on calibration, existing CE estimators either (i) assume access to labeled data from the target domain, often unavailable in practice, or (ii) are derived under a covariate shift assumption. In this work we propose a novel, label-free, consistent CE estimator under *label shift*. Label shift is characterized by changes in the marginal label distribution $p(Y)$, with a constant conditional $p(X|Y)$ distribution between the source and target. We introduce a novel calibration method, called LaSCal, which uses the estimator in conjunction with a post-hoc calibration strategy, to perform unsupervised calibration on the target distribution. Our thorough empirical analysis demonstrates the effectiveness and reliability of the proposed approach across different modalities, model architectures and label shift intensities.

## 1 Introduction

Reliable uncertainty estimation is crucial for predictive models, particularly in safety-critical applications, where decisions based on predictions have significant consequences [Amodei et al., 2016, Kompa et al., 2021]. The *calibration error* (CE) [Naeini et al., 2015, Guo et al., 2017, Vaicenavicius et al., 2019] measures the discrepancy between predicted probabilities and observed class frequencies, indicating the reliability of the model's predictions. When a *calibrated* model predicts an 80% chance of flu, we expect 80 out of 100 patients with similar symptoms to have the flu. Estimating CE and addressing miscalibration typically requires i.i.d. labeled held-out data. However, real-world settings often violate these assumptions: (i) the source (train) may differ from the target (test) distribution, known as *dataset shift* [Quiñonero Candela et al., 2009], leading to a false sense of confidence in the model and suboptimal decision-making [Park et al., 2020]; and (ii) obtaining labeled target data for CE estimation is often unrealistic or prohibitively expensive, e.g., in medical diagnostics during disease outbreaks acquiring labeled patient data is needed, but costly. Thus, traditional post-hoc calibration methods (e.g., temperature scaling [Guo et al., 2017] or isotonic regression [Zadrozny and Elkan, 2002]), which rely on labeled calibration sets in an i.i.d. setting, are not directly applicable.

The two most common types of dataset shift are: (i) *covariate shift*, where the feature distribution changes between the source and target domains, denoted as $p_s(X) \neq p_t(X)$, but the conditional label distribution remains the same, i.e., $p_s(Y|X) = p_t(Y|X)$; and (ii) *label shift*, where the label distribution differs, that is $p_s(Y) \neq p_t(Y)$, while the conditional feature distribution remains the same, i.e., $p_s(X|Y) = p_t(X|Y)$ [Moreno-Torres et al., 2012, Quiñonero Candela et al., 2009][2].

---

[*]Equal contribution.

[2]Label shift corresponds to anti-causal learning: predicting cause $Y$ from effects $X$ [Schölkopf et al., 2012]); e.g., during a pneumonia outbreak $p(Y)$ (flu) rises but the symptoms $p(X|Y)$ (cough given flu) remain the same.

Table 1: Properties of related calibration methods. LaSCal is accuracy preserving, and relies on a consistent CE estimator designed for unsupervised calibration under label-shift.

| Calibration method | Label shift | No target labels | Accuracy preserving | Consistent estimator under label shift assumption |
|---|---|---|---|---|
| TempScal (Source) [Guo et al., 2017] | ✗ | ✗ | ✓ | ✗ |
| CPCS [Park et al., 2020] | ✗ | ✓ | ✓ | ✗ |
| TransCal [Wang et al., 2020] | ✗ | ✓ | ✓ | ✗ |
| HeadToTail [Chen and Su, 2023] | – | ✓ | ✓ | ✗ |
| **LaSCal (Ours)** | ✓ | ✓ | ✓ | ✓ |

Previous methods, such as TransCal [Wang et al., 2020], CPCS [Park et al., 2020], and HeadToTail [Chen and Su, 2023], address calibration under dataset shift, but are specifically designed around the covariate shift assumption. Notably, while HeadToTail also assumes a change in the label distribution between the source and target domains (i.e., $p_s(Y) \neq p_t(Y)$) by using long-tailed source data and balanced target data, it only partially addresses the label shift scenario. As a result, how to effectively estimate and maintain calibration under *label shift* assumption – especially in the absence of target domain labels – remains an open question[3].

To address this gap, we derive *a novel CE estimator of a model facing label shift*, which allows us to reliably estimate CE without requiring labeled target data. Compared to prior work (see Table 1), it is the only label-free, consistent CE estimator under the label shift assumption. We build on ideas from unsupervised domain adaptation, and employ importance weighting to estimate the degree of shift in the target distribution. We utilize current state-of-the-art methods, e.g., ELSA Tian et al. [2023], RLLS Azizzadenesheli et al. [2019], etc., which yield per-class importance weights to account for the label shift. Note that in contrast to our work, these methods focus only on the predictive performance of a model, neglecting the model's calibration altogether.

Furthermore, we propose a novel, accuracy-perserving, post-hoc calibration method, called **LasCal** (Label-Shift Calibration), which utilizes the proposed CE estimator as a loss function. We conduct experiments across a variety of datasets, models, weight estimators, intensities of shift on the target distribution, and imbalance factors of the source distribution to validate its performance. Our results demonstrate that LasCal effectively performs unsupervised calibration on the target domain, yielding better calibrated models, compared to traditional i.i.d. calibration methods, calibration methods designed for covariate shift [Chen and Su, 2023, Park et al., 2020, Wang et al., 2020], and label shift adaptation methods that rely on calibration using a labeled validation (source) set [Alexandari et al., 2020, Wen et al., 2024].

To summarize, we make the following **contributions:**

① We derive the first label-free, consistent calibration error estimator under label shift (§3);

② We propose a post-hoc calibration method, **LaSCal**, which outperforms existing methods in unsupervised calibration tasks in the presence of label shift (§4.1);

③ We analyze the properties of LaSCal, and demonstrate its robustness across various datasets, modalities, model architectures and domain-shift scenarios (§4.2).

Our codebase is released at the following repository: `https://github.com/tpopordanoska/label-shift-calibration`.

## 2 Related Work

**Estimating CE** is a challenging task as it requires estimating an expectation conditioned on a continuous random variable: $\mathbb{E}[Y \mid f(X)]$, where $X$ is the input, $Y$ is a one-hot label, and $f$ is a probabilistic model. The CE is often estimated using binning [Zadrozny and Elkan, 2001, Naeini et al., 2015], i.e., in the *binary* setting, the unit interval $[0, 1]$ is split into intervals (bins) of either equal width [Nguyen and O'Connor, 2015] or equal mass (adaptive binning) [Vaicenavicius et al.,

---

[3]We design a CE estimator under the *label shift* assumption, which involves scenarios where both the source and *target* label distributions may be imbalanced. In contrast, HeadToTail operates under a covariate shift assumption ([Chen and Su, 2023, Section 3.1]), and is applied only in a specific type of label shift.

2019]. Calibration of a *multi-class* model is often quantified via expected CE (ECE) [Naeini et al., 2015], used to assess the so-called top-label (or confidence) calibration [Guo et al., 2017], which only considers the confidence of the predicted class. Class-wise calibration [Kull et al., 2019] is a stronger notion, requiring calibrated scores for each class: $f_k(X)$ is compared with $\mathbb{E}\left[Y_k \mid f_k(X)\right]$ for each class $k$. Canonical calibration [Vaicenavicius et al., 2019, Popordanoska et al., 2022] is the strictest notion, requiring the whole probability vector to be calibrated, i.e., $f(X)$ should match $\mathbb{E}\left[Y \mid f(X)\right]$. In this work, we focus on binary and class-wise CE, estimated using adaptive binning.

Numerous **calibration methods** address neural network miscalibration, falling into two categories: post-hoc and trainable strategies. *Post-hoc methods* adjust the output scores using held-out calibration set. One of the earliest approaches in binary classification is Platt scaling [Platt, 1999], which has been extended to a multi-class setting via matrix, vector and temperature scaling [Guo et al., 2017]. Other approaches include isotonic regression [Zadrozny and Elkan, 2002], ensemble temperature scaling [Zhang et al., 2020], Beta [Kull et al., 2017] and Dirichlet calibration [Kull et al., 2019]. *Trainable methods* incorporate a calibration objective alongside the classification loss [Kumar et al., 2018, Mukhoti et al., 2020, Popordanoska et al., 2022]. All of these strategies focus on a supervised setting, and do not account for dataset shift. Recent studies [Ovadia et al., 2019, Karandikar et al., 2021] have shown that models calibrated with traditional i.i.d. calibration methods lose their calibration under dataset shift. Subsequently, several works address calibration under *covariate shift* assumption: CPCS [Park et al., 2020], TransCal [Wang et al., 2020], and HeadToTail [Chen and Su, 2023], which calibrate on the target domain without labels. In contrast, we focus on the *label shift* setting. Our work introduces a general calibration error estimator under label shift, usable as a training objective in post-hoc and trainable calibration methods. Importantly, post-hoc calibration is done on the unlabeled target data, enhancing performance and reliability compared to standard source data calibration.

**Label shift**, also known as prior probability shift, [Lipton et al., 2018, Azizzadenesheli et al., 2019, Alexandari et al., 2020] is often intertwined with the broader concept of unsupervised domain adaptation [Kouw and Loog, 2021]. Several different methods address label shift: importance re-weighting [Lipton et al., 2018, Azizzadenesheli et al., 2019, Saerens et al., 2002, Tian et al., 2023], kernel mean matching (KMM) [Zhang et al., 2013], and generative adversarial training [Guo et al., 2020]. There are two popular importance re-weighting approaches: one based on maximizing the likelihood function and the other based on inverting a confusion matrix. Saerens et al. [2002] propose an Expectation Maximization (EM) procedure to estimate the class priors shift between the source and target distributions. Importantly, EM does not require retraining or hyperparameter tuning. However, it assumes calibrated predictions, which modern neural networks often lack [Guo et al., 2017]. To address this, hybrid methods combining calibration techniques and domain adaptation methods have been proposed. Alexandari et al. [2020] propose Bias-Corrected Temperature Scaling (BCTS) alongside EM. Lipton et al. [2018] propose Black-Box Shift Learning (BBSL), which estimates the re-weighting coefficients even if the model is poorly calibrated. As an improvement over BBSL, Azizzadenesheli et al. [2019] propose a technique with statistical guarantees: Regularized Learning under Label Shifts (RLLS). They introduce a regularization hyperparameter, addressing the high estimation error of the importance weights in the low target sample regime. Both BBSL and RLLS estimate importance weights from a confusion matrix of a held-out validation set. Both methods cope with label shift when the classifier is miscalibrated, but require model retraining with the importance weights. Recently, Tian et al. [2023] propose a moment-matching framework [Tian et al., 2023] to address label shift, named Efficient Label Shift Adaptation (ELSA). Wen et al. [2024] propose an algorithm called Class Probability Matching with Calibrated Networks (CPMCN), which improves the computational efficiency and empirically outperforms existing methods. Importantly, the goal of these works is to improve the classifier's predictive performance on the label-shifted domain without addressing its calibration. While some methods [Alexandari et al., 2020, Wen et al., 2024] include a calibration step on the labeled validation (source) data to obtain importance weights, they do not calibrate the models on the target domain. In contrast, we propose an approach for target domain calibration without relying on labeled target data. In the absence of target labels, we leverage importance weight estimators to re-weigh the source data.

## 3 Methods

We consider a classification setting where $X \in \mathcal{X} = \mathbb{R}^d$ is the input, and $Y \in \mathcal{Y} = \{0, 1\}^k$ is the one-hot encoded target, with $d$ as the feature space dimensionality, and $k$ the number of classes. The

data consists of: labeled source data $\{(x_i, y_i)\}_{i=1}^{n}$ and unlabeled target data $\{x_i\}_{i=n+1}^{n+m}$. The notation $p_s(\cdot)$ and $p_t(\cdot)$ denotes distributions on the source and target domain, respectively. The support on the target domain is a subset of $\mathcal{Y}$, i.e., the target data does not contain new classes. We use capital letters for unbounded random variables, and lower case letters with subscripts for elements of the data sample. Note that we may still treat elements of the data sample as random variables.

Consider a probabilistic classifier $f \colon \mathcal{X} \to \Delta^k$, where $\Delta^k$ is a $(k-1)$-dimensional probability simplex over $k$ classes, and let $Z = f(X)$ denote the predicted probability distribution for input $X$. We focus on *class-wise calibration error* [Kull et al., 2019, Kumar et al., 2019, Gruber and Buettner, 2022], given by:

$$\mathrm{CWCE}_p(f)^p = \frac{1}{k} \sum_{c=1}^{k} \mathbb{E}\left[|\mathbb{P}(Y_c = 1 \mid Z_c) - Z_c|^p\right], \tag{1}$$

where $Y_c$ denotes the $c^{\text{th}}$ entry in the one-hot label, and $Z_c$ denotes the $c^{\text{th}}$ class confidence score. Since the CE is defined w.r.t. the data distribution, the model's calibration decreases under domain shift, also empirically shown by Ovadia et al. [2019], Karandikar et al. [2021]. However, those works estimate CE on the target shifted data using labels, often unavailable in practice. Thus, we derive an $L_p$ classwise CE estimator for label-free target distribution exhibiting *label shift*.

**Calibration error estimator under label shift.** We consider a label shift: $p_s(Y) \neq p_t(Y)$ and $p_s(X \mid Y) = p_t(X \mid Y)$. We assume the target distribution is absolutely continuous w.r.t. the source; i.e., for every $Y \in \mathcal{Y}$ with $p_t(Y) > 0$, we require $p_s(Y) > 0$ [Lipton et al., 2018] [4]. Assuming access to $n$ labeled source samples, and $m$ unlabeled target samples, we aim to find an estimator:

$$\widehat{\mathrm{CWCE}}_p(f)^p = \frac{1}{k}\frac{1}{m} \sum_{c=1}^{k} \sum_{j=n+1}^{m+n} \left|\widehat{\mathbb{E}_{p_t}[Y_c \mid z_{jc}]} - z_{jc}\right|^p, \tag{2}$$

where the expectations are taken w.r.t. the target, and $z_{jc}$ denotes the $c^{\text{th}}$ entry of the vector $z_j$.

The main challenge is estimating the conditional expectation $\widehat{\mathbb{E}_{p_t}[Y_c \mid z_{jc}]}$ without labels from the target distribution. To re-weigh the source label distribution, we use importance weights $\omega = (\omega_1, \ldots, \omega_k)^T$, where $\omega_i := p_t(Y_c = 1)/p_s(Y_c = 1)$, which we estimate using unsupervised domain adaption methods [Tian et al., 2023, Alexandari et al., 2020, Azizzadenesheli et al., 2019, Lipton et al., 2018]. Then, for the conditional expectation, we have:

$$\mathbb{E}_{p_t}[Y_c \mid Z_c = z_c] = \sum_{y_c \in \{0,1\}} y_c \frac{p_t(Y_c = y_c, Z_c = z_c)}{p_t(Z_c = z_c)} = \frac{p_t(Z_c = z_c \mid Y_c = 1)p_t(Y_c = 1)}{p_t(Z_c = z_c)} \tag{3}$$

$$= \frac{p_s(Z_c = z_c \mid Y_c = 1)p_s(Y_c = 1)\omega_c}{p_t(Z_c = z_c)} \approx \frac{\frac{1}{n}\hat{\omega}_c \sum_{i=1}^{n} \kappa(Z_c = z_c, z_{ic})y_{ic}}{\frac{1}{m} \sum_{i=n+1}^{m+n} \kappa(Z_c = z_c, z_{ic})} \tag{4}$$

where $\omega_c = \frac{p_t(Y_c=1)}{p_s(Y_c=1)}$, $\hat{\omega}_c$ is its empirical estimate, and $\kappa$ is any consistent kernel over its domain [Silverman, 1986]. Under the label shift assumption, we estimate $p_t(Z|Y)$ using $p_s(Z|Y)$ because $p(X|Y)$ remains constant, with $Z = f(X)$ and $f$ being a fixed model. The weights $\hat{\omega}$ are estimated for each $Y \in \mathcal{Y}$ using labeled source and unlabeled target data, along with the model $f$. The error rate of the conditional expectation estimator in Equation (4) is determined by the maximum error rate of its components: the weight $\hat{\omega}_c$ and the ratio estimator. Empirically, we use the RLLS estimator, with an error rate: $\mathcal{O}(n^{-1/2} + m^{-1/2})$ [Azizzadenesheli et al., 2019, Lemma 1] (same as the ratio estimator [Scott and Wu, 1981, Theorem 1]).

**Proposition 3.1** *Given a kernel $\kappa$ consistent over its domain [Silverman, 1986], $\widehat{\mathbb{E}_{p_t}[Y_c \mid Z_c]}$ is a pointwise consistent estimator of $\mathbb{E}_{p_t}[Y_c \mid Z_c]$, that is:*

$$\operatorname*{plim}_{n,m\to\infty} \frac{\frac{1}{n}\hat{\omega}_c \sum_{i=1}^{n} \kappa(Z_c, z_{ic})y_{ic}}{\frac{1}{m} \sum_{i=n+1}^{m+n} \kappa(Z_c, z_{ic})} = \frac{p_t(Z_c = z_c \mid Y_c = 1)p_t(Y_c = 1)}{p_t(Z_c = z_c)} \tag{5}$$

---

[4]The support of the target label distribution should be contained within the source label distribution support.

*Proof sketch.* The proof structure follows [Popordanoska et al., 2022, Proposition 3.2], which demonstrates the pointwise consistency of the ratio estimator. Since the weight estimator is also consistent [Azizzadenesheli et al., 2019, Lemma 1], by the same argument for the product of two convergent sequences of random variables (Proposition 3.2), the conditional expectation estimator is also pointwise consistent.

Plugging Equation (4) back into Equation (2), for CE under label shift we get:

$$\widehat{\mathrm{CWCE}}_p(f)^p = \frac{1}{k}\frac{1}{m}\sum_{c=1}^{k}\sum_{j=n+1}^{m+n}\left|\frac{\frac{1}{n}\hat{\omega}_c\sum_{i=1}^{n}\kappa(z_{jc},z_{ic})y_{ic}}{\frac{1}{m-1}\sum_{\substack{i=n+1\\i\neq j}}^{m+n}\kappa(z_{jc},z_{ic})} - z_{jc}\right|^p . \tag{6}$$

The estimator has values $\in [0,2]$. Since the ratio is pointwise consistent by Proposition 3.1, and following Popordanoska et al. [2022, Proposition 3.5], the CE estimator is consistent for any consistent kernel. Depending on the kernel, the estimator can be differentiable and integrated into post-hoc and trainable calibration methods [5]. We use a binning kernel, returning 1 when $z_{ic}$ and $z_{jc}$ fall in the same bin, and 0 otherwise. The binning estimator yields consistency under well known conditions on the number of bins as a function of the number of observations [Lugosi and Nobel, 1996].

**Unsupervised calibration under label shift.** The CE estimator can be integrated in any post-hoc calibration method, e.g., temperature scaling. In the supervised i.i.d. setting, the optimal temperature $T^*$ is obtained by minimizing the cross entropy loss. In the label shift setting, we propose using LaSCal to find $T^*$ by minimizing the classwise calibration error obtained by the estimator in Equation (6). In particular, let $l_j$ denote the logits corresponding to $z_j$, and $\sigma(\cdot)$ the *softmax* function. We find the optimal temperature $T^*$ as:

$$T^* = \arg\min_{T}\frac{1}{k}\frac{1}{m}\sum_{c=1}^{k}\sum_{j=n+1}^{m+n}\left|\mathbb{E}_{p_t}[\widehat{Y_c \mid \sigma(l_j)/T)_c}] - \sigma(l_j/T)_c\right|^p . \tag{7}$$

## 4 Experiments and Discussion

**Datasets.** To assess the performance of LaSCal for calibrating models facing label shift, we experiment using natural image datasets [Krizhevsky et al., 2009], as well as datasets derived from real-world scenarios [Koh et al., 2021]. In particular, we use the CIFAR-10/100 Long Tail (LT) datasets [Cao et al., 2019], which are simulated from CIFAR [Krizhevsky et al., 2009] with an imbalance factor (IF) defined as a ratio of the number of samples in the most and least prevalent class. We additionally use Wilds [Koh et al., 2021] with different modalities: Camelyon17 [Bandi et al., 2018] and iWildCam [Beery et al., 2021] with images, and Amazon [Ni et al., 2019] with text. Camelyon17 consists of histopathological images of patient lymph node sections with potential metastatic breast cancer. The labels denote whether the central region contains a tumor (binary). iWildCam consists of images from animal traps in the wild, while the labels are different animal species. The Amazon dataset contains review text samples paired with 1-out-of-5 star ratings as labels. Please refer to Appendix A.1 for details about the datasets.

**Metrics.** Unless stated otherwise, we report $L_2$ calibration error (CE) $\times 100$ [Kumar et al., 2019], fix the number of bins to 15, and use adaptive binning strategy [Vaicenavicius et al., 2019]). In multi-class settings, we report the sum of per-class CE. We perform a bootstrap procedure, i.e., repeatedly resampling with replacement and estimating CE on each subset, and we report the mean and standard deviation of the estimates. In the reliability diagrams, we report $L_1$ top-label CE (ECE).

**Models.** For the experiments we conduct on CIFAR-10/100 we use ResNet [He et al., 2016] models initialized from scratch with different depths (20, 32, 56, 110). For experiments on iWildCam we report results with a standard ResNet-50 [He et al., 2016], two ViT-large transformer-based models [Dosovitskiy et al., 2020] (with an image resolution of 224 or 384), and Swin-Large [Liu et al., 2021] (all pre-trained on ImageNet). For experiments on Amazon, we use pre-trained transformer-based models: BERT [Devlin et al., 2018], RoBERTa [Liu et al., 2019], DistillBert [Sanh et al., 2019] and DistillRoBERTa [Sanh et al., 2019]. For the experiments on Camelyon17, we use a ResNet-50 pre-trained on ImageNet. Please refer to Appendix A.2 for implementation details.

---

[5]Popordanoska et al. [2022] and Zhang et al. [2020] proposed Dirichlet and Triweight kernels, respectively.

## 4.1 Calibration under label shift

We compare the performance of LaSCal as a method for post-hoc calibration of a model trained on a source distribution against several (state-of-the-art) baselines. We compare against: (i) **Uncal**: uncalibrated model trained on source data; (ii) **TempScal** calibrated model using temperature scaling on source data; (iii) **CPCS** [Park et al., 2020], **TransCal** [Wang et al., 2020], and **HeadToTail** Chen and Su [2023]: calibrated models using adapted versions of TempScal, derived under covariate shift assumption. HeadToTail additionally assumes long-tailed source data, and balanced target data. (iv) **EM-BCTS** [Alexandari et al., 2020] and **CPMCN** [Wen et al., 2024]: methods for label shift adaptation, where a calibration step is performed on the source data prior to obtaining the importance weights. Note that TempScal relies only on labeled source data, while the other baselines also incorporate unlabeled target data. Additionally, in Appendix A.3 we include other common, post-hoc, source-domain calibration methods: vector scaling (**VectScal**) [Guo et al., 2017], an ensemble method designed to improve the expressivity of TempScal, abbreviated as **EnsTempScal** [Zhang et al., 2020], and one-versus-all isotonic regression (**IROvA**) [Zadrozny and Elkan, 2002].

We train ResNet models on the CIFAR-10/100 LT variants, and use IF $= 10$, i.e., the least frequent class is subsampled to 10% of the original size, while the target is balanced. The iWildCam and Amazon datasets have an i.i.d. validation set, serving as our source distribution, and an i.i.d. test set, to which we apply label shift and use it as our target distribution. We use the i.i.d. test set to ensure that the input distribution $p(X)$ remains the same. On iWildCam, we select the 20 most frequent classes from the target dataset. On both iWildCam and Amazon, we obtain a uniform target distribution by subsampling each class, based on the frequency of the least frequent class.

*Performance of LaSCal across various modalities, datasets and models.* In Table 2, we report CE on the label-shifted (balanced) target domain before and after calibration with various post-hoc methods. We observe that LaSCal either achieves a lower macro-averaged CE across models compared to other methods, or performs on par with the top-performing method, irrespective of the input modality. Compared to the second best method – EM-BCTS – where the calibration is performed on the labeled source data, the proposed LaSCal is explicitly derived for *unsupervised* calibration on a label-shifted *target* distribution. Notably, LaSCal significantly outperforms other baselines on CIFAR-100, where around 50% of the classes contain less than 30 source data points (see Figure 5 in Appendix A.1). This highlights LaSCal's effectiveness even in low data regimes[6]. In Appendix A.3, we report accuracy, additional experiments using other IFs on CIFAR-10/100, and provide results for the scenario where the source is balanced and the target is long-tailed, as commonly studied in related works [Tian et al., 2023, Alexandari et al., 2020, Lipton et al., 2018, Azizzadenesheli et al., 2019].

*Performance of LaSCal compared to temperature scaling using labels.* In Fig. 1 (Left), we evaluate how closely LaSCal (without labels) approaches the performance of temperature scaling applied on the target distribution using labels, referred to as TempScal (Target), which serves as a competitive baseline. For comparison, we also include temperature scaling applied on the source distribution, referred to as TempScal (Source), representing a lower reference point for the method's performance. Throughout these experiments, we keep the input distribution $p(X)$ fixed. Across different models on iWildCam and Amazon, we observe that LaSCal performs favorably relative to TempScal (Source) and closes the gap with TempScal (Target), demonstrating its effectiveness in unsupervised calibration.

*Label shift with changing input distribution.* We consider an alternative setting where both the label $p(Y)$ and input $p(X)$ distributions change, which is common in real-world applications[7]. We investigate this scenario by using the out-of-distribution (OOD) test sets of Amazon and iWildCam ($p(X)$ changes), to which we apply label shift, and use them as our target distribution. The iWildCam OOD test set contains images of camera traps from locations absent from the source distribution, with variation in illumination, camera angle, background, vegetation, etc. [Beery et al., 2021]. The Amazon OOD test set contains reviews from users outside of the source distribution. In Fig. 1 (Right), we report the CE after post-hoc calibration using temperature scaling on the source distribution (TempScal), using HeadToTail[8], and LaSCal. We observe that both HeadToTail and LaSCal signifi-

---

[6]We noticed that the optimal temperature obtained by LaSCal for CIFAR-100 is considerably higher than related methods. We hypothesize the discrepancy arises from the optimization process.

[7]For example, in medical diagnosis, the model might be facing label shift because of a pandemic, while also dealing with patient data from different demographics between training and testing.

[8]We chose HeadToTail because it is explicitly designed to address the setting where both the input and label distributions change between source and target domains.

Table 2: CE on label-shifted target domain before and after calibration with various post-hoc methods. LaSCal performs unsupervised calibration by minimizing CE on the unlabeled target distribution and either outperforms, or performs competitively with the other methods in all scenarios.

| Model | Uncal | TempScal | CPCS | TransCal | HeadToTail | EM-BCTS | CPMCN | LaSCal |
|---|---|---|---|---|---|---|---|---|
| ***CIFAR-10-LT (IF=10)*** | | | | | | | | |
| ResNet-20 | $8.87_{\pm0.38}$ | $4.55_{\pm0.18}$ | $4.61_{\pm0.17}$ | $5.05_{\pm0.20}$ | $4.71_{\pm0.14}$ | $\mathbf{3.77}_{\pm0.12}$ | $3.79_{\pm0.11}$ | $4.44_{\pm0.17}$ |
| ResNet-32 | $10.45_{\pm0.41}$ | $5.03_{\pm0.24}$ | $5.18_{\pm0.21}$ | $6.59_{\pm0.32}$ | $4.87_{\pm0.15}$ | $4.99_{\pm0.24}$ | $5.19_{\pm0.21}$ | $\mathbf{4.81}_{\pm0.17}$ |
| ResNet-56 | $11.25_{\pm0.31}$ | $4.82_{\pm0.18}$ | $5.10_{\pm0.18}$ | $6.96_{\pm0.20}$ | $4.75_{\pm0.14}$ | $\mathbf{4.41}_{\pm0.11}$ | $4.42_{\pm0.12}$ | $4.57_{\pm0.15}$ |
| ResNet-110 | $11.89_{\pm0.35}$ | $5.12_{\pm0.18}$ | $5.12_{\pm0.17}$ | $7.51_{\pm0.26}$ | $4.78_{\pm0.14}$ | $\mathbf{4.40}_{\pm0.12}$ | $4.43_{\pm0.13}$ | $4.70_{\pm0.16}$ |
| *Macro average* | 10.62 | 4.88 | 5.00 | 6.53 | 4.78 | **4.39** | 4.46 | 4.63 |
| ***CIFAR-100-LT (IF = 10)*** | | | | | | | | |
| ResNet-20 | $65.66_{\pm0.23}$ | $24.61_{\pm0.24}$ | $24.12_{\pm0.23}$ | $48.15_{\pm0.29}$ | $19.93_{\pm0.21}$ | $25.01_{\pm0.21}$ | $25.02_{\pm0.24}$ | $\mathbf{5.62}_{\pm0.08}$ |
| ResNet-32 | $71.16_{\pm0.24}$ | $28.29_{\pm0.24}$ | $24.84_{\pm0.25}$ | $57.55_{\pm0.24}$ | $28.37_{\pm0.23}$ | $26.17_{\pm0.21}$ | $24.76_{\pm0.20}$ | $\mathbf{5.80}_{\pm0.07}$ |
| ResNet-56 | $72.24_{\pm0.21}$ | $29.71_{\pm0.22}$ | $25.03_{\pm0.24}$ | $59.27_{\pm0.28}$ | $29.72_{\pm0.27}$ | $26.33_{\pm0.23}$ | $24.53_{\pm0.22}$ | $\mathbf{5.88}_{\pm0.07}$ |
| ResNet-110 | $72.80_{\pm0.21}$ | $31.55_{\pm0.25}$ | $26.51_{\pm0.25}$ | $60.52_{\pm0.27}$ | $31.58_{\pm0.27}$ | $28.22_{\pm0.24}$ | $26.49_{\pm0.22}$ | $\mathbf{6.19}_{\pm0.08}$ |
| *Macro average* | 70.47 | 28.54 | 25.13 | 56.37 | 27.40 | 26.43 | 25.20 | **5.87** |
| ***Amazon Reviews*** | | | | | | | | |
| RoBERTa | $11.44_{\pm0.79}$ | $4.91_{\pm0.31}$ | $4.20_{\pm0.39}$ | $4.36_{\pm0.36}$ | $4.36_{\pm0.36}$ | $2.72_{\pm0.35}$ | $\mathbf{1.36}_{\pm0.17}$ | $3.66_{\pm0.29}$ |
| DistillRoBERTa | $17.82_{\pm0.98}$ | $5.21_{\pm0.45}$ | $3.60_{\pm0.31}$ | $7.75_{\pm0.60}$ | $2.90_{\pm0.21}$ | $\mathbf{2.13}_{\pm0.28}$ | $2.81_{\pm0.23}$ | $2.72_{\pm0.23}$ |
| BERT | $27.33_{\pm0.98}$ | $7.75_{\pm0.55}$ | $4.34_{\pm0.39}$ | $16.98_{\pm0.98}$ | $\mathbf{3.62}_{\pm0.30}$ | $3.95_{\pm0.40}$ | $9.32_{\pm0.54}$ | $3.72_{\pm0.34}$ |
| DistillBERT | $22.18_{\pm1.14}$ | $6.54_{\pm0.51}$ | $3.94_{\pm0.32}$ | $11.89_{\pm0.75}$ | $3.43_{\pm0.29}$ | $3.41_{\pm0.36}$ | $5.48_{\pm0.34}$ | $\mathbf{3.40}_{\pm0.28}$ |
| *Macro average* | 19.19 | 6.10 | 4.02 | 10.25 | 3.58 | **3.05** | 4.74 | 3.38 |
| ***iWildCam*** | | | | | | | | |
| ResNet50 | $18.44_{\pm0.74}$ | $16.38_{\pm0.61}$ | $\mathbf{11.52}_{\pm0.93}$ | $13.81_{\pm0.48}$ | $15.53_{\pm0.56}$ | $15.84_{\pm0.57}$ | $19.43_{\pm0.69}$ | $13.07_{\pm0.45}$ |
| Swin-Large | $22.07_{\pm0.84}$ | $17.39_{\pm0.66}$ | $17.57_{\pm0.69}$ | $16.42_{\pm0.49}$ | $16.55_{\pm0.57}$ | $16.81_{\pm0.63}$ | $18.03_{\pm0.62}$ | $\mathbf{15.43}_{\pm0.54}$ |
| ViT-Large | $17.94_{\pm0.71}$ | $16.78_{\pm0.66}$ | $20.24_{\pm0.80}$ | $13.64_{\pm0.48}$ | $16.53_{\pm0.66}$ | $24.83_{\pm1.31}$ | $19.33_{\pm0.80}$ | $\mathbf{13.07}_{\pm0.50}$ |
| Vit-Large (384) | $18.99_{\pm0.86}$ | $18.78_{\pm0.80}$ | $21.92_{\pm0.96}$ | $\mathbf{14.81}_{\pm0.52}$ | $17.92_{\pm0.77}$ | $19.78_{\pm0.73}$ | $20.74_{\pm0.72}$ | $17.27_{\pm0.66}$ |
| *Macro average* | 19.36 | 17.33 | 17.81 | **14.67** | 16.63 | 19.31 | 19.38 | 14.71 |

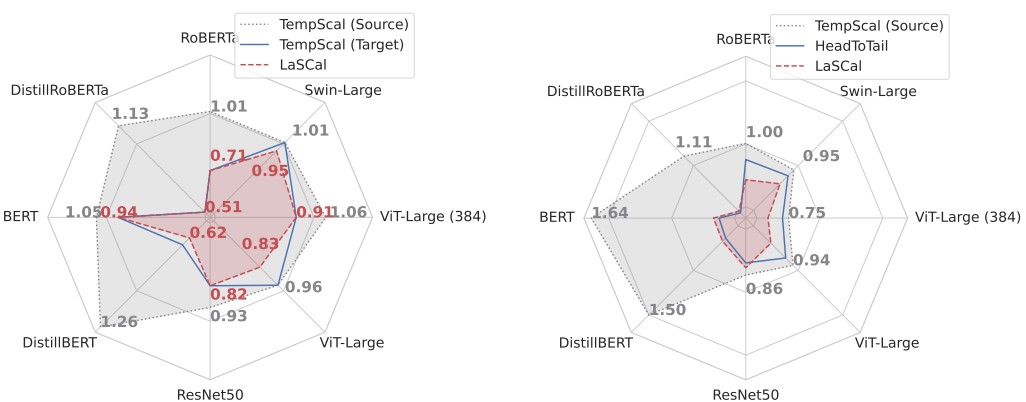

Figure 1: **Left.** Comparison of LaSCal with temperature scaling on the source distribution (TempScal Source) or target distribution (TempScal Target), using labels. **Right.** CE after post-hoc calibration on iWildCam and Amazon when the target distribution exhibits both label and covariate shift w.r.t. the source. We report CE normalized by the number of classes (5 for Amazon and 20 for iWildCam) for illustration purposes. Lower numbers are better.

cantly outperform TempScal across all models. Furthermore, despite the more challenging setting, we observe that LaSCal performs on par or outperforms the HeadToTail method.

*Top-label calibration.* While classwise calibration is central to our analysis, top-label calibration remains a popular approach in related works. To gain insights about this notion of calibration, we present reliability diagrams in Fig. 2 for DistillRoBERTa trained on Amazon, allowing us to visually assess the calibration quality across confidence levels. We report CE of (a) an uncalibrated model, (b) after applying temperature scaling on the source domain, (c) after label-shift adaption using EM-BCTS, and (d) after applying temperature scaling using LaSCal. The blue bars indicate

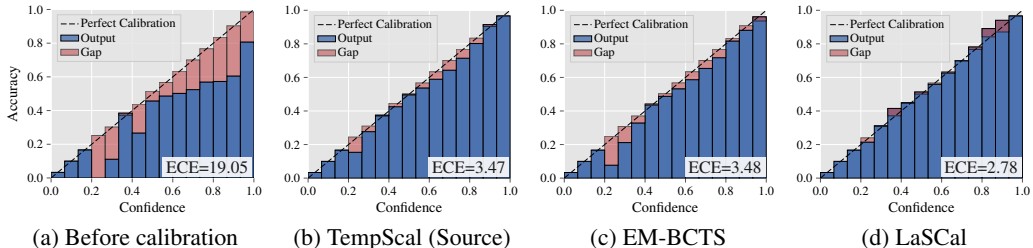

| (a) Before calibration | (b) TempScal (Source) | (c) EM-BCTS | (d) LaSCal |

Figure 2: Reliability diagrams on Amazon using DistillRoBERTa before and after calibration. TempScal (Source) and EM-BCTS perform IID calibration on the source distribution. LaSCal calibrates the model on the unlabeled target distribution. We report $L_1$ top-label CE in the bottom right corner.

the accuracy per bin, and the red bars represent the gap of each bin to perfect calibration, i.e., the difference between accuracy and confidence for a given bin (darker shades signify under-confidence, while brighter red colors denote over-confidence). We observe that LaSCal provides better calibration (i.e., lower ECE) compared to other baselines, as also confirmed by the reported $L_1$ top-label CE in the bottom right corner of each plot. Furthermore, note that while the classwise CE values of EM-BCTS are better than LaSCal (Table 2), the diagrams reveal that top-label confidence scores obtained with LaSCal are favorable compared to EM-BCTS in most bins.

Building on this, we adapt our approach to top-label calibration, and we include the adapted estimator, along with empirical comparisons with competing calibration methods in Appendix B. The results on Amazon and iWildCam further validate the effectiveness of LaSCal, which continues to outperform other methods, demonstrating its superior calibration capabilities in the top-label setting.

## 4.2 Empirical analysis of the estimator's properties

**Robustness analysis.** Using the Camelyon17 dataset, we conduct a series of experiments to assess the impact of various factors on the performance of the CE estimator, and report the results in Fig. 3. We partition the original training set as the source distribution, and we use the i.i.d. validation set to form a target set with varying label distribution shifts. We chose this dataset because (i) the application is both realistic and safety-critical; (ii) the dataset is balanced across source and target, enabling us to alter both distributions as per the setting we are trying to verify; (iii) the problem is binary, allowing us to study the estimator properties on a simple problem. For all experiments, we use a ResNet-50 model pre-trained on ImageNet, subsequently fine-tuned on the Camelyon17 dataset.

Across the experiments, we construct the train and validation sets, sampled from the source distribution, by keeping all negative samples and sampling a portion of the positives. Unless stated otherwise, we report results by sampling $20\%$ of the positive samples for training: i.e., $5:1$ ratio of negative to positive points. We compare the estimated CE values (without labels) to the ground truth (with labels), across different experimental scenarios, designed to assess the impact of a change in the data distribution and the sample size on the CE estimation. To account for the variability in data resampling, we average the results across 10 iterations. For each iteration, we apply bootstrap sampling and compute the mean and variance of the estimated CE. Finally, we report the overall mean and standard deviation (depicted as shaded region in the plot) across all iterations.

*Impact of data distribution changes.* In Fig. 3a, we investigate the effect of increasing the label shift intensity of the target distribution. We impose a constraint such that the size of the source and target distribution is the same ($n = m$), and we systematically shift the target by modifying the ratio of negative to positive samples: $5:1, 4:1, ..., 1:1, ..., 1:4$. Therefore, in the most favorable scenario ($5:1$), the source and target distribution are the same (no label shift), while in the extreme $1:4$, we have 4 times as many positive samples in the target data (which could occur, e.g., during a disease outbreak). We observe that the estimated CE closely follows the ground truth, even in the most extreme case. Furthermore, we observe that the variance increases with the intensity of the shift, indicating greater uncertainty and reducing the confidence one should have in the CE estimates. In Fig. 3b we analyze the effect of changing the ratio of source to target samples ($n:m$), while keeping the total number of points ($n + m$) constant. The source distribution has a $5:1$ ratio of negative to positive points, while the target is balanced. We observe that the estimator achieves

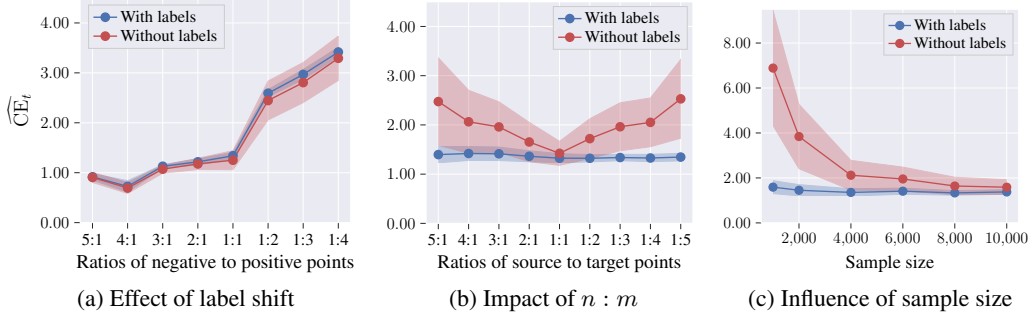

| (a) Effect of label shift | (b) Impact of $n:m$ | (c) Influence of sample size |

Figure 3: Robustness analysis. We report mean and standard deviation over multiple iterations of resampling the data. $5:1$ denotes "no shift", and the severity increases from left to right. LaSCal generalizes well to a wide range of shifts, ratios of source to target samples, and sample sizes.

optimal performance when the source and target data sizes are equal ($n:m$). As the ratio increases, the estimated values begin to diverge slightly from the ground truth. However, the true values lie within the estimator's standard deviation in most cases, demonstrating that even in more extreme settings – $5\times$ more source than target samples – our estimator yields reliable CE estimates.

*Impact of data size.* In Fig. 3c, we constrain that $n = m$, and we incrementally vary the sample size from $1,000$ to $10,000$ samples. In practice, low data regimes are common where annotated data is costly to obtain. We observe that the CE estimates deviate from the ground truth the most when using the fewest data points (i.e., 1000), and improve as the data quantity increases. Furthermore, our estimator has a positive bias w.r.t. ground truth in the small data regime, indicating that the estimator tends to be conservative in that setting. Such tendency is preferable in this context, as it prevents mistakenly reporting good performance of a model on the basis of not enough samples. In essence, our method errs on the side of caution, ensuring reliability even in data-constrained environments.

**Method analysis.** We further investigate (i) how different weight estimation methods influence the estimator's effectiveness, and (ii) to what extent our CE estimates (without labels) deviate from the values obtained using labeled target data.

*Impact of the weight estimation method.* Our estimator relies on the availability of per-class importance weights, which can be obtained using domain adaptation methods, such as ELSA [Tian et al., 2023], RLLS [Azizzadenesheli et al., 2019] and BBSL [Lipton et al., 2018]. In Fig. 4 (Left), we compare the performance of these methods when integrated in our estimator. We observe that RLLS emerges as the most favorable compared to the others, providing reasonable importance weights in all settings. See Appendix A.4 for experiments involving other weight estimators.

*Performance evaluation.* In Fig. 4 (Right), we investigate the CE measured by our estimator, compared to the ground-truth CE (obtained using labels from the target domain). Additionally, we report the CE on the source domain as a reference point. We observe that the calibration error increases from the source to target domain when the model is facing label shift. Importantly, our estimator ($\widehat{CE_t}$) effectively closes the gap to the ground-truth ($CE_t$), consistently yielding accurate CE estimates across models on Amazon and iWildCam.

## 5 Discussion and Conclusion

In this work, we addressed the problem of estimating CE of an *unlabeled* target distribution under *label shift*. We observe that prior state-of-the-art methods Wang et al. [2020], Chen and Su [2023] only address CE estimation under the covariate shift assumption: $p_s(X) \neq p_t(X)$ and $p_s(Y|X) = p_t(Y|X)$; while, to the best of our knowledge, we propose the first CE estimator under the label shift assumption: $p_s(Y) \neq p_t(Y)$ and $p_s(X|Y) = p_t(X|Y)$. We demonstrated that it yields CE estimates closely reflecting the ground truth. Furthermore, we showcase that our estimator can be successfully used as a post-hoc calibration method – LaSCal – for unsupervised model calibration on a target distribution. Overall, our experiments indicate that LaSCal can effectively minimize the CE across

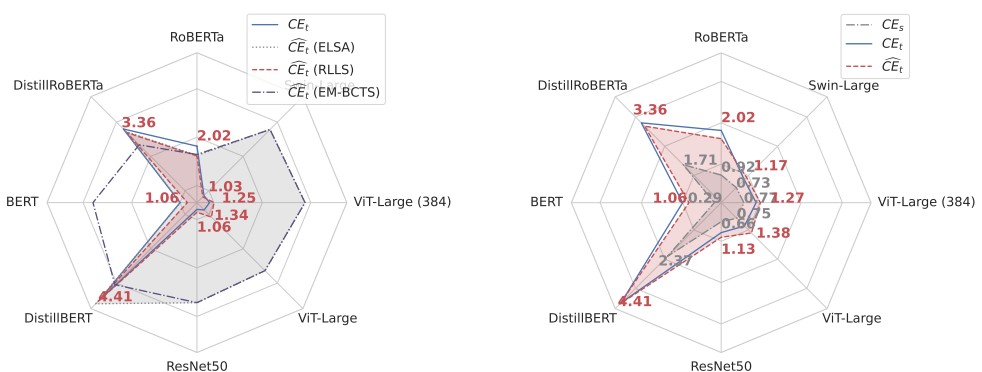

Figure 4: **Left.** Impact of the weight estimation method on our estimator. **Right.** Our estimator ($\widehat{CE}_t$) effectively closes the gap to the ground truth ($CE_t$). We report CE normalized by the number of classes (5 for Amazon and 20 for iWildCam) for illustration purposes.

different intensities of label shift and various modalities. Finally, we analyze the properties of the estimator and contribute towards a nuanced understanding of its strengths and weaknesses.

**Limitations.** First, LaSCal is specifically designed to address label-shift, however, other types of dataset-shift are equally important, e.g., *covariate shift* [Shimodaira, 2000]. Note that our experiments in §4.1 shed light on the scenario where the models encounter both label shift and shift in $p(X)$, and we observed favorable results compared to prior work. Nevertheless, we consider designing consistent CE estimators under covariate shift a crucial direction for future work. Second, the estimator we propose is dependent on how well the importance weights reflect the ground truth between the classes, which we obtain from current methods [Tian et al., 2023, Azizzadenesheli et al., 2019, Alexandari et al., 2020]. Expectedly, we inherit some limitations of such methods: if certain classes are under-represented, the importance weights could be unreliable. However, our experiments in Appendix A.4 showcase that our estimator consistently improves as the weights become more accurate. Third, the estimator requires a sufficient number of data samples, e.g., 4000 samples in Figure 3c, to accurately estimate the calibration error. In severely data-scarce settings, this requirement may limit potential applications. However, the error rate of our estimator is $(n^{-1/2} + m^{-1/2})$, which is the same as the weight estimation methods (see Azizzadenesheli et al. [2019, Lemma 1] for the RLLS method, and Garg et al. [2020, page 8, top paragraph] for the EM-BCTS method). Therefore, the data requirement is not unique to our method, but rather it is common across all weight estimation-based approaches.

**Broader impact.** Our proposed approach effectively reduces CE under label shift, and allows for a more comprehensive and realistic evaluation of model calibration. We consider the ethical risks to be essentially the same as for any probabilistic classifier. Overall, we consider this paper a significant step toward improving the model's robustness and reliability, crucial for safety-critical applications.

# Acknowledgments

This research received funding from the Research Foundation - Flanders (FWO) through project number S001421N, the Flemish Government under the "Onderzoeksprogramma Artificiële Intelligentie (AI) Vlaanderen" programme, and KULeuven Methusalem.

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

# Supplementary Material
# LaSCal: Label-Shift Calibration without target labels

The Supplementary material is organized as follows:

- Details about the datasets we use for the experiments (§A.1).
- Implementation details (§A.2)
- Additional experiments with LaSCal (§A.3).
- The proposed estimator with different importance weight estimators (§A.4).

## A    Experiments

In this section, we include more details about the used datasets and training procedures. We also report additional experiments to evaluate the performance of our proposed method using different importance weight estimators.

### A.1    Details about the datasets

We report statistics for all datasets in Table 3.

**CIFAR-10/100** [Krizhevsky et al., 2009, Cao et al., 2019]. Using the CIFAR datasets we examine two different types of label shift. In **Task A** the source distribution is long-tailed, whereas the target is uniform. In **Task B** the source distribution is uniform, and the target is long-tailed. The CIFAR-10/100 Long-Tail datasets are simulated from CIFAR-10/100, respectively, with different imbalance factors (IF). The IF controls the ratio between the number of samples in the most frequent and the least frequent class. For example, an imbalance factor of 10 indicates that the least frequent class appears 10 times less than the most frequent one. In Figure 5 we show the number of target images per class on the long-tailed CIFAR-10/100 with imbalance factors ranging from 1.25 to 100.

In **Task A** we keep the target distribution unchanged (i.e., balanced across classes), and we resample the source distribution with various IF. In the main paper, we presented a setting with source IF $= 10$ in Table 2. Additional results using different imbalance factors induced on the source distribution are given in Tables 11 – 16. In **Task B** the models are trained on the original (balanced) CIFAR datasets, and in Table 10 in Appendix A.3 we report the performance of our CE estimator on label-shifted target distribution with various imbalance factors.

Table 3: Statistics for all datasets used in the paper. Note that we report the original number of classes and samples of the datasets we use.

| Dataset | Modality | Num. classes | Train samples | Val samples | Test samples |
|---------|----------|--------------|---------------|-------------|--------------|
| CIFAR-10 | Images | 10 | 40,000 | 10,000 | 10,000 |
| CIFAR-100 | Images | 100 | 40,000 | 10,000 | 10,000 |
| Camelyon17 | Images | 2 | 302436 | 33560 | – |
| iWildCam | Images | 182 | 129809 | 7314 | 8154 |
| Amazon | Text | 5 | 245502 | 46950 | 46950 |

**Camelyon17** [Bandi et al., 2018] consists of $96 \times 96$ whole-side images (WSI) of breast-cancer metastases in lymph node sections collected from hospitals in the Netherlands. In each WSI, the tumor regions are annotated manually by pathologists. The labels indicate whether the central $32 \times 32$ region contains a tumor. As Camelyon17 contains only a training and validation set—drawn from the same (source) distribution—both of which are balanced across the positive and negative class, we perform the following: (i) We use the validation set as testing dataset, which we convert to our desired target distribution by resampling the positive class; (ii) From the training dataset, we allocate a validation dataset with the same size as the testing dataset. Then, we subsample the validation dataset the same way as we subsample the training dataset, so that both are effectively drawn from the same (source) distribution (e.g., used in the ablation studies in Section 4.3).

**iWildCam** [Beery et al., 2021] consists of images obtained from animal camera traps (i.e., heat or motion-activated static cameras placed in the wild) which are set in countries in different parts of

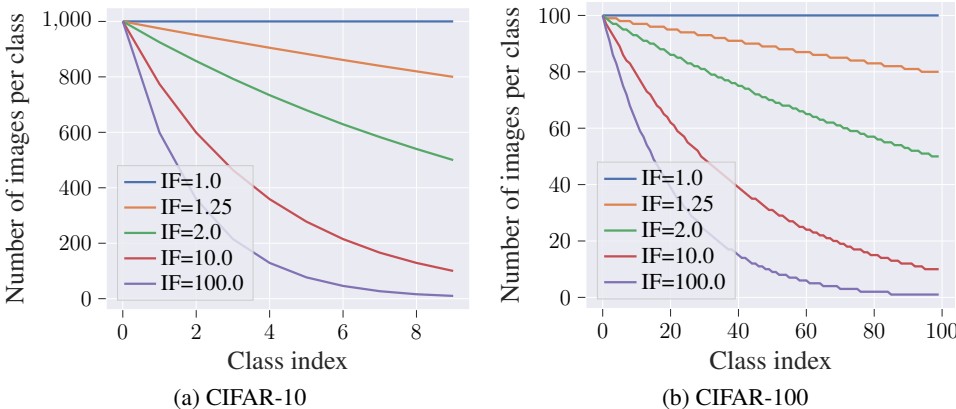

Figure 5: Number of target samples per class in simulated long-tail CIFAR-10/100 datasets with different imbalance factors (IF).

the world. The label of each image is one of the 182 animal species. The training and validation set features the same long-tail distribution among classes. As testing data from the target distribution, we use the original test dataset and we perform the following: (i) we keep only the 20 most frequent classes; (ii) we resample the dataset such that classes follow a uniform distribution (thus representing the target label-shifted distribution), with an imposed minimal frequency of 84 samples per testing class. Therefore, the number of samples in the test dataset (target distribution) is 1680, while the validation set (source distribution) contains 6003 samples.

**Amazon** [Ni et al., 2019] consists of texts which represent user reviews, while the label is 1-out-of-5 score of the review. The training and validation set (the source distribution) follows the same long-tail distribution among classes. As the testing dataset is also long-tail, we resample each class based on the frequency of the least frequent class, yielding a test set following a uniform distribution of classes, representing the target, where each class appears 527 times. Therefore, the number of samples in the test dataset (target distribution) is 2860.

### A.2 Implementation details

We conduct all experiments on consumer-grade GPUs, that is, all experiments can be conducted on a single Nvidia 3090. We use PyTorch [Paszke et al., 2019] for all deep-learning-based implementations. Below we provide further information about the training procedure for each of the datasets, along with implementation details of the weight estimators we use.

**CIFAR-10/100 (Long-Tail).** We keep the same training procedure for both CIFAR-10/100 and their long-tail variants. Namely, we train all models with stochastic gradient descent (SGD) for 200 epochs, with a peak learning rate of 0.1, linearly warmed up for the first 10% of the training, and then decreased to 0.0 until the end. We apply weight decay of 0.0005, and clip the gradients when their norm exceeds 5.0. During training, we augment the images by applying random horizontal flips.

**WILDS datasets (Camelyon17, iWildCam and Amazon).** We keep the same training procedure across all WILDS datasets, with the only difference across datasets being the data augmentation and the used models. Namely, we train all models with AdamW Loshchilov and Hutter [2017] for 10 epochs, with a peak learning rate of 0.0005, linearly warmed up for the first 10% of the training and then decreased to 0.0 until the end. We apply weight decay of 0.001, and clip the gradients when their norm exceeds 5.0. During training, for models trained on Amazon we do not apply any data augmentation on the input text, while for models trained on Camelyon17 and iWildCam we obtain a random crop of the image with size $224 \times 224$, perform horizontal flipping, and apply color jitter with parameters: brightness=(0.6, 1.4), contrast=(0.6, 1.4), saturation=(0.6, 1.4). During testing, we obtain a single crop with size $224 \times 224$ from the center of the image. For all ImageNet pre-trained models we use Timm [Wightman et al., 2019] (Camelyon17 and iWildCam), while for all pre-trained language models, we use HuggingFace transformers [Wolf et al., 2019]. On Camelyon17 and iWildCam, we train a diverse set of transformer based-models which are pre-trained on ImageNet:

ResNet50 [He et al., 2016], ViT-Large and ViT-Large with input resolution of 384 [Dosovitskiy et al., 2020], and Swin-Large [Liu et al., 2021]. On Amazon, we train different transformer-based language models: BERT (bert-base-uncased) [Devlin et al., 2018], D-BERT (distilbert-base-uncased) [Sanh et al., 2019], RoBERTa (roberta-base) [Liu et al., 2019], and D-RoBERTa (distilroberta-base) [Sanh et al., 2019].

**Weight estimators.** Our proposed method relies on estimating importance weights using techniques from the unsupervised domain adaptation literature. Most of the weight estimators (RLLS [Aziz-zadenesheli et al., 2019], BBSL [Lipton et al., 2018], EM-BCTS [Saerens et al., 2002]) that we use are implemented in `https://github.com/kundajelab/abstention`. For the ELSA [Tian et al., 2023] method, we used the original implementation provided by the authors. In some of our settings, we detected issues with the weight estimation methods, prompting us to set a minimal value of the confidence scores to $1 \times 10^{-15}$ for EM-BCTS, $1 \times 10^{-3}$ for ELSA, and $1 \times 10^{-2}$ for BBSL, in order to get a reasonable estimate of the weights in most cases. We also encountered issues with the BBSL method on the iWildCam dataset, due to the source distribution containing 0-frequency classes. RLLS consistently delivered the most accurate and stable weight estimations, thus, we report our main results using this method. Note that in certain experiments, some of the importance weight estimation methods (e.g., BBSL) yield poor estimates, resulting in abnormal values for the calibration error. However, addressing these issues is beyond the scope of this paper, as they are specific to the weight estimation methods, and not with our CE estimator.

### A.3 Additional experiments with LaSCal as a post-hoc calibration method

In Table 4 we report additional performance evaluation of LaSCal compared to other post-hoc calibration strategies on CIFAR-10/100, where the model is trained on a long-tail source distribution obtained with various imbalance factors (IF). We report CE on the balanced target distribution. The missing values of the HeadToTail method are due to singular matrix error encountered when running the method, using the original code of the paper.

Table 4: CE on label-shifted target domain before and after calibration with various post-hoc methods.

| Model | Uncal | TempScal | VectScal | EnsTempScal | IROvA | CPCS | TransCal | HeadToTail | LaSCal |
|---|---|---|---|---|---|---|---|---|---|
| *CIFAR-10-LT (IF = 5)* | | | | | | | | | |
| ResNet-20 | $8.38_{\pm0.24}$ | $4.71_{\pm0.16}$ | $4.68_{\pm0.15}$ | $4.96_{\pm0.16}$ | $5.55_{\pm0.20}$ | $4.90_{\pm0.20}$ | $4.67_{\pm0.17}$ | $4.60_{\pm0.13}$ | $4.41_{\pm0.12}$ |
| ResNet-32 | $10.38_{\pm0.25}$ | $5.41_{\pm0.16}$ | $5.34_{\pm0.16}$ | $5.78_{\pm0.17}$ | $6.23_{\pm0.20}$ | $5.63_{\pm0.19}$ | $5.99_{\pm0.17}$ | $4.68_{\pm0.14}$ | $4.69_{\pm0.15}$ |
| ResNet-56 | $11.86_{\pm0.31}$ | $5.37_{\pm0.16}$ | $5.47_{\pm0.17}$ | $5.17_{\pm0.18}$ | $6.53_{\pm0.27}$ | $5.65_{\pm0.16}$ | $7.19_{\pm0.18}$ | $4.76_{\pm0.12}$ | $4.74_{\pm0.13}$ |
| ResNet-110 | $13.19_{\pm0.30}$ | $5.69_{\pm0.18}$ | $5.77_{\pm0.17}$ | $5.33_{\pm0.14}$ | $6.80_{\pm0.27}$ | $5.70_{\pm0.20}$ | $9.37_{\pm0.31}$ | – | $4.84_{\pm0.13}$ |
| *CIFAR-100-LT (IF = 5)* | | | | | | | | | |
| ResNet-20 | $65.16_{\pm0.25}$ | $26.45_{\pm0.24}$ | $27.50_{\pm0.28}$ | $27.55_{\pm0.26}$ | $31.14_{\pm0.24}$ | $26.26_{\pm0.28}$ | $47.48_{\pm0.25}$ | $21.69_{\pm0.25}$ | $5.97_{\pm0.07}$ |
| ResNet-32 | $71.32_{\pm0.20}$ | $28.75_{\pm0.29}$ | $28.84_{\pm0.22}$ | $29.54_{\pm0.28}$ | $31.08_{\pm0.24}$ | $27.66_{\pm0.25}$ | $57.39_{\pm0.24}$ | $28.67_{\pm0.26}$ | $6.19_{\pm0.08}$ |
| ResNet-56 | $73.07_{\pm0.18}$ | $33.18_{\pm0.26}$ | $30.27_{\pm0.27}$ | $32.57_{\pm0.29}$ | $31.51_{\pm0.23}$ | $29.03_{\pm0.27}$ | $61.13_{\pm0.26}$ | $33.12_{\pm0.27}$ | $6.47_{\pm0.07}$ |
| ResNet-110 | $73.99_{\pm0.19}$ | $35.23_{\pm0.28}$ | $30.74_{\pm0.25}$ | $34.29_{\pm0.27}$ | $32.15_{\pm0.22}$ | $29.58_{\pm0.28}$ | $62.93_{\pm0.27}$ | $35.27_{\pm0.27}$ | $6.61_{\pm0.08}$ |
| *CIFAR-10-LT (IF = 2)* | | | | | | | | | |
| ResNet-20 | $9.09_{\pm0.14}$ | $5.55_{\pm0.13}$ | $5.48_{\pm0.13}$ | $5.96_{\pm0.13}$ | $6.30_{\pm0.14}$ | $5.87_{\pm0.13}$ | $5.28_{\pm0.12}$ | – | $4.69_{\pm0.12}$ |
| ResNet-32 | $11.02_{\pm0.24}$ | $6.29_{\pm0.13}$ | $6.05_{\pm0.14}$ | $6.85_{\pm0.12}$ | $7.22_{\pm0.21}$ | $6.61_{\pm0.11}$ | $6.58_{\pm0.14}$ | – | $5.39_{\pm0.13}$ |
| ResNet-56 | $13.86_{\pm0.38}$ | $6.84_{\pm0.12}$ | $6.81_{\pm0.12}$ | $6.71_{\pm0.13}$ | $9.52_{\pm0.63}$ | $7.14_{\pm0.13}$ | $9.18_{\pm0.21}$ | – | $5.43_{\pm0.11}$ |
| ResNet-110 | $13.63_{\pm0.58}$ | $7.23_{\pm0.20}$ | $7.19_{\pm0.29}$ | $6.84_{\pm0.31}$ | $8.28_{\pm0.24}$ | $7.55_{\pm0.22}$ | $9.28_{\pm0.26}$ | – | $6.44_{\pm0.23}$ |
| *CIFAR-100-LT (IF = 2)* | | | | | | | | | |
| ResNet-20 | $61.35_{\pm0.27}$ | $32.12_{\pm0.30}$ | $32.82_{\pm0.25}$ | $33.10_{\pm0.29}$ | $37.73_{\pm0.26}$ | $32.23_{\pm0.29}$ | $42.40_{\pm0.29}$ | $27.98_{\pm0.26}$ | $6.64_{\pm0.08}$ |
| ResNet-32 | $70.53_{\pm0.21}$ | $33.85_{\pm0.27}$ | $34.66_{\pm0.29}$ | $34.44_{\pm0.31}$ | $37.08_{\pm0.24}$ | $32.83_{\pm0.28}$ | $56.77_{\pm0.26}$ | $29.74_{\pm0.26}$ | $6.99_{\pm0.07}$ |
| ResNet-56 | $74.42_{\pm0.17}$ | $37.08_{\pm0.30}$ | $36.31_{\pm0.30}$ | $36.92_{\pm0.28}$ | $37.47_{\pm0.26}$ | $33.52_{\pm0.29}$ | $63.66_{\pm0.27}$ | $37.10_{\pm0.29}$ | $7.44_{\pm0.08}$ |
| ResNet-110 | $75.50_{\pm0.17}$ | $41.37_{\pm0.27}$ | $38.95_{\pm0.31}$ | $41.02_{\pm0.32}$ | $40.02_{\pm0.26}$ | $36.06_{\pm0.33}$ | $66.23_{\pm0.24}$ | $41.32_{\pm0.28}$ | $7.55_{\pm0.08}$ |

In Table 5 we report additional performance evaluation of LaSCal against traditional, i.i.d, post-hoc calibration methods. Note that VectScal, EnsTempScal, and IROvA are not specifically designed for label-shift scenarios and rely solely on labeled source data. In contrast, LaSCal is tailored for situations where label shift occurs, leveraging both the labeled source data and unlabeled target data to perform calibration. This enables LaSCal to adapt to changes in the class distribution between the source and target domains, providing better calibration under such conditions, as reflected in the reported results.

In Figure 6 and Figure 7 we show reliability diagrams for CIFAR-10 using ResNet-20, and Amazon using DistillBERT, respectively, before and after calibration. Similar to the results presented in the main text, we observe that LaSCal obtains lowest ECE.

Table 5: CE on label-shifted target domain before and after calibration with various post-hoc methods.

| Model | Uncal | VectScal | EnsTempScal | IROvA | LaSCal |
|---|---|---|---|---|---|
| *CIFAR-10-LT (IF=10)* | | | | | |
| ResNet-20 | $8.87_{\pm0.38}$ | $5.30_{\pm0.19}$ | $4.72_{\pm0.20}$ | $5.19_{\pm0.24}$ | $4.44_{\pm0.17}$ |
| ResNet-32 | $10.45_{\pm0.41}$ | $5.13_{\pm0.19}$ | $5.43_{\pm0.22}$ | $5.78_{\pm0.26}$ | $4.81_{\pm0.17}$ |
| ResNet-56 | $11.25_{\pm0.31}$ | $5.14_{\pm0.17}$ | $4.69_{\pm0.18}$ | $6.01_{\pm0.25}$ | $4.57_{\pm0.15}$ |
| ResNet-110 | $11.89_{\pm0.35}$ | $5.20_{\pm0.20}$ | $4.97_{\pm0.17}$ | $5.89_{\pm0.29}$ | $4.70_{\pm0.16}$ |
| *Macro average* | 10.62 | 5.19 | 4.95 | 5.72 | 4.63 |
| *CIFAR-100-LT (IF = 10)* | | | | | |
| ResNet-20 | $65.66_{\pm0.23}$ | $25.91_{\pm0.22}$ | $25.37_{\pm0.20}$ | $28.75_{\pm0.24}$ | $5.62_{\pm0.08}$ |
| ResNet-32 | $71.16_{\pm0.24}$ | $26.14_{\pm0.22}$ | $27.84_{\pm0.23}$ | $27.76_{\pm0.24}$ | $5.80_{\pm0.07}$ |
| ResNet-56 | $72.24_{\pm0.21}$ | $26.67_{\pm0.25}$ | $28.95_{\pm0.27}$ | $28.52_{\pm0.21}$ | $5.88_{\pm0.07}$ |
| ResNet-110 | $72.80_{\pm0.21}$ | $28.36_{\pm0.26}$ | $30.96_{\pm0.26}$ | $29.96_{\pm0.19}$ | $6.19_{\pm0.08}$ |
| *Macro average* | 70.47 | 26.77 | 28.28 | 28.75 | 5.87 |
| *Amazon Reviews* | | | | | |
| RoBERTa | $11.44_{\pm0.79}$ | $5.48_{\pm0.45}$ | $4.86_{\pm0.43}$ | $4.88_{\pm0.42}$ | $3.66_{\pm0.29}$ |
| DistillRoBERTa | $17.82_{\pm0.98}$ | $5.84_{\pm0.40}$ | $5.27_{\pm0.41}$ | $4.80_{\pm0.40}$ | $2.72_{\pm0.23}$ |
| BERT | $27.33_{\pm0.98}$ | $8.47_{\pm0.52}$ | $7.74_{\pm0.59}$ | $7.02_{\pm0.45}$ | $3.62_{\pm0.30}$ |
| DistillBERT | $22.18_{\pm1.14}$ | $7.36_{\pm0.47}$ | $6.52_{\pm0.54}$ | $6.40_{\pm0.46}$ | $3.40_{\pm0.28}$ |
| *Macro average* | 19.19 | 6.79 | 6.10 | 5.78 | 3.38 |
| *iWildCam* | | | | | |
| ResNet50 | $18.44_{\pm0.74}$ | $21.10_{\pm1.13}$ | $16.61_{\pm0.58}$ | $16.07_{\pm0.70}$ | $13.07_{\pm0.45}$ |
| Swin-Large | $22.07_{\pm0.84}$ | $20.89_{\pm0.99}$ | $17.36_{\pm0.62}$ | $16.49_{\pm0.69}$ | $15.43_{\pm0.54}$ |
| ViT-Large | $17.94_{\pm0.71}$ | $20.96_{\pm0.95}$ | $17.07_{\pm0.69}$ | $16.87_{\pm0.86}$ | $13.07_{\pm0.50}$ |
| Vit-Large (384) | $18.99_{\pm0.86}$ | $21.64_{\pm1.07}$ | $18.69_{\pm0.75}$ | $16.49_{\pm0.77}$ | $17.27_{\pm0.66}$ |
| *Macro average* | 19.36 | 21.15 | 17.43 | 16.48 | 14.71 |

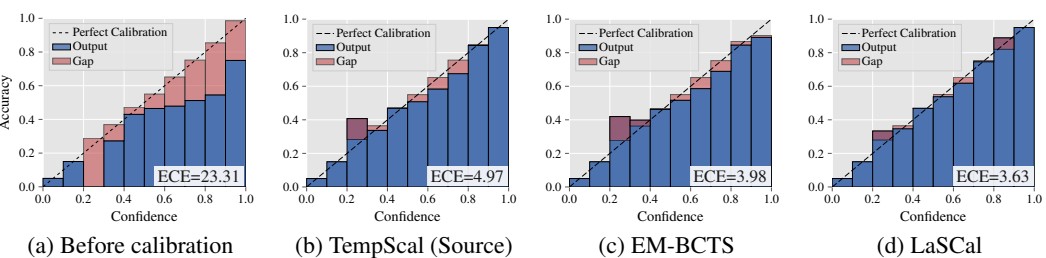

| (a) Before calibration | (b) TempScal (Source) | (c) HeadToTail | (d) LaSCal |
|---|---|---|---|

Figure 6: Reliabiliy diagrams on CIFAR-10 using ResNet-20 before and after calibration.

| (a) Before calibration | (b) TempScal (Source) | (c) EM-BCTS | (d) LaSCal |
|---|---|---|---|

Figure 7: Reliability diagrams on Amazon using DistillRoBERTa before and after calibration.

As some of the post-hoc calibration strategies are not accuracy-preserving, in Tables 6 to 8 we report accuracy for the uncalibrated model, as well as for vector scaling, and isotonic regression. We observe that the change in accuracy is negligible in most cases.

Table 6: Accuracy of CIFAR-10/100 before and after calibration with non accuracy-perserving calibration methods. The error bars represent 95% CI.

| | CIFAR-10-LT (IF=10) | | | CIFAR-100-LT (IF=10) | | |
|---|---|---|---|---|---|---|
| | Uncal | VectScal | IROvA | Uncal | VectScal | IROvA |
| ResNet-20 | $78.24_{\pm0.81}$ | $77.18_{\pm0.82}$ | $77.34_{\pm0.82}$ | $44.81_{\pm0.97}$ | $44.00_{\pm0.97}$ | $43.91_{\pm0.97}$ |
| ResNet-32 | $80.74_{\pm0.77}$ | $80.74_{\pm0.77}$ | $80.61_{\pm0.77}$ | $47.73_{\pm0.98}$ | $46.55_{\pm0.98}$ | $46.68_{\pm0.98}$ |
| ResNet-56 | $81.71_{\pm0.76}$ | $81.09_{\pm0.77}$ | $81.25_{\pm0.77}$ | $47.18_{\pm0.98}$ | $46.76_{\pm0.98}$ | $46.30_{\pm0.98}$ |
| ResNet-110 | $81.29_{\pm0.76}$ | $81.22_{\pm0.77}$ | $81.09_{\pm0.77}$ | $49.78_{\pm0.98}$ | $49.22_{\pm0.98}$ | $49.35_{\pm0.98}$ |

Table 7: Accuracy on iWildCam.

| | Uncal | VectScal | IROvA |
|---|---|---|---|
| ResNet50 | $70.83_{\pm2.17}$ | $68.33_{\pm2.22}$ | $68.69_{\pm2.22}$ |
| Swin-Large | $70.83_{\pm2.17}$ | $67.02_{\pm2.25}$ | $69.52_{\pm2.20}$ |
| ViT-Large | $66.96_{\pm2.25}$ | $63.99_{\pm2.30}$ | $64.52_{\pm2.29}$ |
| Vit-Large (384) | $69.64_{\pm2.20}$ | $66.61_{\pm2.26}$ | $67.86_{\pm2.23}$ |

Table 8: Accuracy on Amazon.

| | Uncal | VectScal | IROvA |
|---|---|---|---|
| RoBERTa | $56.26_{\pm1.82}$ | $55.00_{\pm1.82}$ | $54.48_{\pm1.83}$ |
| DistillRoBERTa | $56.99_{\pm1.81}$ | $55.63_{\pm1.82}$ | $55.91_{\pm1.82}$ |
| BERT | $53.36_{\pm1.83}$ | $52.48_{\pm1.83}$ | $52.13_{\pm1.83}$ |
| DistillBERT | $55.03_{\pm1.82}$ | $53.99_{\pm1.83}$ | $53.64_{\pm1.83}$ |

In Table 9 we show the full comparison of LaSCal and other post-hoc calibration methods, in a setting where additionally to the label shift, the input distribution changes. Across most settings, our method achieves best or second best calibration error.

Table 9: Label shift with changing input distribution $p(X)$

| Model | Uncal | TempScal | VectScal | EnsTempScal | IROvA | CPCS | TransCal | HeadToTail | LaSCal (TS) |
|---|---|---|---|---|---|---|---|---|---|
| **Amazon Reviews** | | | | | | | | | |
| RoBERTa | $11.91_{\pm0.44}$ | $5.02_{\pm0.26}$ | $5.49_{\pm0.24}$ | $4.97_{\pm0.24}$ | $4.96_{\pm0.24}$ | $4.16_{\pm0.21}$ | $4.38_{\pm0.23}$ | $4.38_{\pm0.24}$ | $\mathbf{3.58_{\pm0.16}}$ |
| DistillRoBERTa | $19.30_{\pm0.72}$ | $5.53_{\pm0.27}$ | $6.11_{\pm0.24}$ | $5.56_{\pm0.31}$ | $5.00_{\pm0.23}$ | $3.37_{\pm0.20}$ | $8.33_{\pm0.38}$ | $\mathbf{2.35_{\pm0.13}}$ | $2.44_{\pm0.13}$ |
| BERT | $28.05_{\pm0.63}$ | $8.19_{\pm0.32}$ | $8.42_{\pm0.26}$ | $8.08_{\pm0.31}$ | $6.96_{\pm0.28}$ | $3.87_{\pm0.22}$ | $17.77_{\pm0.56}$ | $\mathbf{3.12_{\pm0.17}}$ | $3.34_{\pm0.20}$ |
| DistillBERT | $25.44_{\pm0.60}$ | $7.50_{\pm0.35}$ | $8.26_{\pm0.33}$ | $7.43_{\pm0.38}$ | $7.07_{\pm0.30}$ | $4.16_{\pm0.20}$ | $14.00_{\pm0.46}$ | $\mathbf{3.19_{\pm0.18}}$ | $3.36_{\pm0.18}$ |
| *Macro average* | 21.18 | 6.56 | 7.07 | 6.51 | 6.0 | 3.89 | 11.12 | 3.26 | **3.18** |
| **iWildCam** | | | | | | | | | |
| ResNet50 | $21.43_{\pm0.48}$ | $17.28_{\pm0.38}$ | $16.69_{\pm0.36}$ | $17.28_{\pm0.34}$ | $3.75_{\pm0.27}$ | $36.63_{\pm0.74}$ | $\mathbf{13.51_{\pm0.26}}$ | $15.35_{\pm0.33}$ | $16.13_{\pm0.35}$ |
| Swin-Large | $26.75_{\pm0.59}$ | $18.84_{\pm0.40}$ | $22.26_{\pm0.52}$ | $18.80_{\pm0.39}$ | $4.85_{\pm4.90}$ | $38.65_{\pm0.89}$ | $17.49_{\pm0.34}$ | $17.20_{\pm0.35}$ | $\mathbf{13.86_{\pm0.28}}$ |
| Vit-Large | $16.05_{\pm0.35}$ | $15.03_{\pm0.31}$ | $14.48_{\pm0.32}$ | $15.08_{\pm0.28}$ | $14.53_{\pm0.39}$ | $22.43_{\pm0.54}$ | $11.95_{\pm0.26}$ | $14.07_{\pm0.26}$ | $\mathbf{11.76_{\pm0.26}}$ |
| Vit-Large (384) | $20.03_{\pm0.42}$ | $19.02_{\pm0.37}$ | $19.18_{\pm0.39}$ | $19.05_{\pm0.33}$ | $4.30_{\pm0.45}$ | $21.00_{\pm0.40}$ | $\mathbf{15.29_{\pm0.33}}$ | $17.72_{\pm0.33}$ | $15.91_{\pm0.35}$ |
| *Macro average* | 21.07 | 17.54 | 18.15 | 17.55 | 6.86 | 29.68 | 14.56 | 16.09 | **14.41** |

Additionally, several related works Tian et al. [2023], Alexandari et al. [2020], Lipton et al. [2018], Azizzadenesheli et al. [2019] predominantly focus on an alternative type of label shift, where the source is balanced (i.e., the classes have equal frequency), while the target follows a long-tail distribution. We report results for such settings in Table 10, where we induce label shift on the target data with imbalance factors with magnitudes: 10 and 100. The accuracy remains unaffected by the induced label shift. As before, we perform experiments with ResNet models of varying depths to verify that our findings generalize across models of different complexities. The results on both datasets reveal that the estimator yields reliable CE values in the absence of labeled target data, irrespective of the IF intensity.

### A.4 Effect of different importance weight estimators

In this section, we report additional experiments to assess the effectiveness of our proposed approach using various importance weight estimation methods: RLLS, ELSA, EM-BCTS, and BBSL.

In Tables 11 – 16 we report accuracy, ground truth CE (using labels) and estimated CE using different importance weight estimators. The models are trained on CIFAR-10/100-LT. Each table corresponds to different IF imposed on the source distribution, and we report CE with different $L_p$ norms: $L_1$ or $L_2$. For all experiments, the target distribution is uniform. The subscripts $s$ and $t$ denote the source and target distributions, respectively.

When encountering a less severe label shift: Table 12 (IF $= 5$) and Table 13 (IF $= 2$), we observe a comparable performance across all weight estimators. However, under more pronounced label shift: Table 11 (IF $= 10$), in several settings we encounter issues with ELSA, EM-BCTS and BBSL

Table 10: Performance evaluation of our estimator on a label-shifted target distribution using different imbalance factors, with models trained on balanced CIFAR-10/100. Across both shifts, LaSCal ($\widehat{\mathrm{CE}_t}$) yields accurate estimates compared to the ground truth (with labels), and effectively handles even the severe case with IF = 100. The error bars represent 95% CI for Acc, and average standard deviation across classes for CE.

| | | | IF=10.0 | | IF=100.0 | |
|---|---|---|---|---|---|---|
| Model | $\mathrm{Acc}_t$ | $\mathrm{CE}_t$ | $\mathrm{CE}_t$ | $\widehat{\mathrm{CE}_t}$ | $\mathrm{CE}_t$ | $\widehat{\mathrm{CE}_t}$ |
| ***CIFAR-10*** | | | | | | |
| ResNet-20 | $87.22_{\pm0.65}$ | $8.76_{\pm0.12}$ | $7.95_{\pm0.32}$ | $8.70_{\pm0.40}$ | $10.77_{\pm0.50}$ | $12.13_{\pm0.72}$ |
| ResNet-32 | $88.47_{\pm0.63}$ | $10.72_{\pm0.21}$ | $9.36_{\pm0.42}$ | $9.71_{\pm0.52}$ | $10.57_{\pm0.41}$ | $10.88_{\pm0.53}$ |
| ResNet-56 | $88.53_{\pm0.62}$ | $10.02_{\pm0.14}$ | $8.47_{\pm0.37}$ | $8.82_{\pm0.48}$ | $10.10_{\pm0.55}$ | $10.78_{\pm0.53}$ |
| ResNet-110 | $90.00_{\pm0.59}$ | $13.74_{\pm0.33}$ | $10.42_{\pm0.62}$ | $10.36_{\pm0.60}$ | $10.45_{\pm0.58}$ | $10.83_{\pm0.58}$ |
| ***CIFAR-100*** | | | | | | |
| ResNet-20 | $61.19_{\pm0.96}$ | $59.15_{\pm0.26}$ | $57.08_{\pm0.42}$ | $57.54_{\pm0.48}$ | $53.83_{\pm0.78}$ | $53.38_{\pm0.83}$ |
| ResNet-32 | $62.71_{\pm0.95}$ | $67.24_{\pm0.24}$ | $65.16_{\pm0.46}$ | $65.77_{\pm0.42}$ | $61.10_{\pm0.80}$ | $60.20_{\pm0.81}$ |
| ResNet-56 | $65.07_{\pm0.93}$ | $74.59_{\pm0.19}$ | $72.58_{\pm0.35}$ | $73.55_{\pm0.45}$ | $68.67_{\pm0.80}$ | $69.51_{\pm0.84}$ |
| ResNet-110 | $65.96_{\pm0.93}$ | $76.04_{\pm0.18}$ | $73.83_{\pm0.35}$ | $74.27_{\pm0.38}$ | $67.72_{\pm0.75}$ | $68.15_{\pm0.83}$ |

methods, resulting in abnormal CE values. In contrast, the RLLS method yields stable and reliable values across all settings. The CE estimates obtained using RLLS often closely align with those of the CE estimator that utilizes ground truth weights, denoted as $\boldsymbol{\omega}^*$.

Table 11: Comparison of different importance weight estimators. The source is obtained with an IF = 10. We measure $L_2$ CE. The abnormal values on CIFAR-100-LT with BBSL and ELSA are due to issues with the weight estimators in this setting.

| Model | $\mathrm{Acc}_s$ | $\mathrm{Acc}_t$ | $\mathrm{CE}_s$ | $\mathrm{CE}_t$ | $\widehat{\mathrm{CE}}_t(\boldsymbol{\omega}^*)$ | $\widehat{\mathrm{CE}}_t(\hat{\boldsymbol{\omega}})$ RLLS | $\widehat{\mathrm{CE}}_t(\hat{\boldsymbol{\omega}})$ ELSA | $\widehat{\mathrm{CE}}_t(\hat{\boldsymbol{\omega}})$ EM-BCTS | $\widehat{\mathrm{CE}}_t(\hat{\boldsymbol{\omega}})$ BBSL |
|---|---|---|---|---|---|---|---|---|---|
| ***CIFAR-10-LT*** | | | | | | | | | |
| ResNet-20 | $83.10_{\pm1.15}$ | $78.24_{\pm0.81}$ | $8.19_{\pm0.41}$ | $8.86_{\pm0.33}$ | $9.26_{\pm0.29}$ | $8.93_{\pm0.33}$ | $9.06_{\pm0.31}$ | $8.57_{\pm0.48}$ | $9.01_{\pm0.40}$ |
| ResNet-32 | $85.48_{\pm1.08}$ | $80.74_{\pm0.77}$ | $9.18_{\pm0.50}$ | $10.45_{\pm0.40}$ | $11.75_{\pm0.40}$ | $12.16_{\pm0.38}$ | $12.03_{\pm0.39}$ | $11.20_{\pm0.47}$ | $12.19_{\pm0.39}$ |
| ResNet-56 | $85.38_{\pm1.08}$ | $81.71_{\pm0.76}$ | $9.87_{\pm0.54}$ | $11.24_{\pm0.32}$ | $11.67_{\pm0.33}$ | $11.71_{\pm0.24}$ | $11.62_{\pm0.34}$ | $11.19_{\pm0.49}$ | $11.77_{\pm0.30}$ |
| ResNet-110 | $84.94_{\pm1.10}$ | $81.29_{\pm0.76}$ | $10.06_{\pm0.58}$ | $11.95_{\pm0.38}$ | $12.28_{\pm0.35}$ | $12.21_{\pm0.33}$ | $12.18_{\pm0.33}$ | $11.69_{\pm0.41}$ | $12.26_{\pm0.35}$ |
| ***CIFAR-100-LT*** | | | | | | | | | |
| ResNet-20 | $52.24_{\pm1.57}$ | $44.81_{\pm0.97}$ | $61.28_{\pm0.38}$ | $65.67_{\pm0.24}$ | $65.63_{\pm0.28}$ | $63.97_{\pm0.28}$ | $96.61_{\pm0.54}$ | $63.09_{\pm0.27}$ | $134078.52_{\pm1462.04}$ |
| ResNet-32 | $53.48_{\pm1.57}$ | $47.73_{\pm0.98}$ | $65.99_{\pm0.38}$ | $71.24_{\pm0.26}$ | $71.42_{\pm0.28}$ | $70.03_{\pm0.14}$ | $83.05_{\pm0.31}$ | $69.06_{\pm0.17}$ | $464.12_{\pm4.54}$ |
| ResNet-56 | $54.21_{\pm1.57}$ | $47.18_{\pm0.98}$ | $66.12_{\pm0.37}$ | $72.21_{\pm0.18}$ | $72.46_{\pm0.16}$ | $71.33_{\pm0.34}$ | $1048.32_{\pm11.97}$ | $69.89_{\pm0.21}$ | $12147.40_{\pm393.55}$ |
| ResNet-110 | $56.58_{\pm1.56}$ | $49.78_{\pm0.98}$ | $68.87_{\pm0.45}$ | $72.87_{\pm0.16}$ | $73.03_{\pm0.17}$ | $72.11_{\pm0.17}$ | $77.34_{\pm0.24}$ | $70.72_{\pm0.15}$ | $82.88_{\pm0.22}$ |

Table 12: Comparison of different importance weight estimators. The source is obtained with an IF = 5. We measure $L_2$ CE.

| Model | $\mathrm{Acc}_s$ | $\mathrm{Acc}_t$ | $\mathrm{CE}_s$ | $\mathrm{CE}_t$ | $\widehat{\mathrm{CE}}_t(\boldsymbol{\omega}^*)$ | $\widehat{\mathrm{CE}}_t(\hat{\boldsymbol{\omega}})$ RLLS | $\widehat{\mathrm{CE}}_t(\hat{\boldsymbol{\omega}})$ ELSA | $\widehat{\mathrm{CE}}_t(\hat{\boldsymbol{\omega}})$ EM-BCTS | $\widehat{\mathrm{CE}}_t(\hat{\boldsymbol{\omega}})$ BBSL |
|---|---|---|---|---|---|---|---|---|---|
| ***CIFAR-10-LT*** | | | | | | | | | |
| ResNet-20 | $84.77_{\pm0.99}$ | $82.77_{\pm0.74}$ | $7.21_{\pm0.23}$ | $8.40_{\pm0.22}$ | $8.93_{\pm0.23}$ | $8.89_{\pm0.24}$ | $9.07_{\pm0.23}$ | $8.75_{\pm0.32}$ | $8.92_{\pm0.22}$ |
| ResNet-32 | $86.68_{\pm0.93}$ | $83.47_{\pm0.73}$ | $8.03_{\pm0.31}$ | $10.36_{\pm0.26}$ | $10.40_{\pm0.23}$ | $10.44_{\pm0.24}$ | $10.41_{\pm0.22}$ | $10.17_{\pm0.29}$ | $10.47_{\pm0.24}$ |
| ResNet-56 | $86.03_{\pm0.95}$ | $84.17_{\pm0.72}$ | $9.59_{\pm0.70}$ | $11.82_{\pm0.30}$ | $11.74_{\pm0.24}$ | $11.68_{\pm0.28}$ | $11.77_{\pm0.29}$ | $11.53_{\pm0.29}$ | $11.73_{\pm0.28}$ |
| ResNet-110 | $86.44_{\pm0.94}$ | $85.04_{\pm0.70}$ | $9.99_{\pm0.71}$ | $13.17_{\pm0.36}$ | $13.52_{\pm0.31}$ | $13.54_{\pm0.30}$ | $13.58_{\pm0.28}$ | $13.35_{\pm0.35}$ | $13.47_{\pm0.31}$ |
| ***CIFAR-100-LT*** | | | | | | | | | |
| ResNet-20 | $52.92_{\pm1.39}$ | $50.39_{\pm0.98}$ | $62.49_{\pm0.32}$ | $65.19_{\pm0.30}$ | $65.16_{\pm0.19}$ | $65.74_{\pm0.23}$ | $67.02_{\pm0.24}$ | $64.19_{\pm0.27}$ | $68.56_{\pm0.19}$ |
| ResNet-32 | $55.52_{\pm1.39}$ | $50.64_{\pm0.98}$ | $68.51_{\pm0.45}$ | $71.24_{\pm0.25}$ | $71.35_{\pm0.18}$ | $72.05_{\pm0.25}$ | $73.16_{\pm0.19}$ | $70.17_{\pm0.19}$ | $74.04_{\pm0.23}$ |
| ResNet-56 | $56.59_{\pm1.38}$ | $53.33_{\pm0.98}$ | $70.27_{\pm0.32}$ | $73.16_{\pm0.14}$ | $73.17_{\pm0.25}$ | $73.60_{\pm0.23}$ | $75.48_{\pm0.17}$ | $71.96_{\pm0.16}$ | $75.62_{\pm0.20}$ |
| ResNet-110 | $57.26_{\pm1.38}$ | $54.16_{\pm0.98}$ | $71.39_{\pm0.30}$ | $73.85_{\pm0.20}$ | $74.10_{\pm0.19}$ | $74.05_{\pm0.15}$ | $74.95_{\pm0.24}$ | $72.45_{\pm0.21}$ | $76.00_{\pm0.16}$ |

Table 13: Comparison of different importance weight estimators. The source is obtained with an IF $= 2$. We measure $L_2$ CE.

| Model | $\text{Acc}_s$ | $\text{Acc}_t$ | $\text{CE}_s$ | $\text{CE}_t$ | $\widehat{\text{CE}}_t(\boldsymbol{\omega}^*)$ | $\widehat{\text{CE}}_t(\hat{\boldsymbol{\omega}})$ RLLS | $\widehat{\text{CE}}_t(\hat{\boldsymbol{\omega}})$ ELSA | $\widehat{\text{CE}}_t(\hat{\boldsymbol{\omega}})$ EM-BCTS | $\widehat{\text{CE}}_t(\hat{\boldsymbol{\omega}})$ BBSL |
|---|---|---|---|---|---|---|---|---|---|
| **_CIFAR-10-LT_** | | | | | | | | | |
| ResNet-20 | $86.74_{\pm 0.78}$ | $85.78_{\pm 0.68}$ | $5.27_{\pm 0.33}$ | $9.05_{\pm 0.14}$ | $9.48_{\pm 0.16}$ | $9.53_{\pm 0.15}$ | $9.59_{\pm 0.14}$ | $9.55_{\pm 0.17}$ | $9.54_{\pm 0.14}$ |
| ResNet-32 | $86.94_{\pm 0.78}$ | $86.82_{\pm 0.66}$ | $5.59_{\pm 0.35}$ | $11.00_{\pm 0.25}$ | $11.14_{\pm 0.21}$ | $11.03_{\pm 0.25}$ | $11.01_{\pm 0.22}$ | $11.01_{\pm 0.23}$ | $11.06_{\pm 0.21}$ |
| ResNet-56 | $88.41_{\pm 0.74}$ | $87.93_{\pm 0.64}$ | $7.84_{\pm 0.53}$ | $13.80_{\pm 0.33}$ | $13.85_{\pm 0.34}$ | $13.94_{\pm 0.39}$ | $13.94_{\pm 0.35}$ | $13.93_{\pm 0.33}$ | $13.93_{\pm 0.35}$ |
| ResNet-110 | $88.33_{\pm 0.74}$ | $87.51_{\pm 0.65}$ | $7.71_{\pm 0.52}$ | $13.60_{\pm 0.55}$ | $13.83_{\pm 0.53}$ | $13.99_{\pm 0.48}$ | $13.93_{\pm 0.51}$ | $13.79_{\pm 0.51}$ | $14.01_{\pm 0.47}$ |
| **_CIFAR-100-LT_** | | | | | | | | | |
| ResNet-20 | $56.81_{\pm 1.15}$ | $56.71_{\pm 0.97}$ | $60.41_{\pm 0.27}$ | $61.43_{\pm 0.23}$ | $61.17_{\pm 0.21}$ | $61.80_{\pm 0.18}$ | $61.76_{\pm 0.18}$ | $61.15_{\pm 0.29}$ | $123.74_{\pm 1.72}$ |
| ResNet-32 | $58.43_{\pm 1.14}$ | $58.59_{\pm 0.97}$ | $70.46_{\pm 0.26}$ | $70.48_{\pm 0.22}$ | $70.53_{\pm 0.28}$ | $71.24_{\pm 0.15}$ | $71.33_{\pm 0.21}$ | $70.22_{\pm 0.25}$ | $71.66_{\pm 0.15}$ |
| ResNet-56 | $60.42_{\pm 1.13}$ | $59.96_{\pm 0.96}$ | $73.93_{\pm 0.18}$ | $74.49_{\pm 0.17}$ | $74.30_{\pm 0.19}$ | $74.86_{\pm 0.16}$ | $74.74_{\pm 0.20}$ | $74.07_{\pm 0.17}$ | $74.86_{\pm 0.22}$ |
| ResNet-110 | $62.88_{\pm 1.12}$ | $61.99_{\pm 0.95}$ | $74.93_{\pm 0.20}$ | $75.45_{\pm 0.08}$ | $75.39_{\pm 0.20}$ | $75.74_{\pm 0.20}$ | $75.88_{\pm 0.15}$ | $75.07_{\pm 0.16}$ | $75.68_{\pm 0.18}$ |

Table 14: Comparison of different importance weight estimators. The source is obtained with an IF $= 10$. We measure $L_1$ CE.

| Model | $\text{Acc}_s$ | $\text{Acc}_t$ | $\text{CE}_s$ | $\text{CE}_t$ | $\widehat{\text{CE}}_t(\boldsymbol{\omega}^*)$ | $\widehat{\text{CE}}_t(\hat{\boldsymbol{\omega}})$ RLLS | $\widehat{\text{CE}}_t(\hat{\boldsymbol{\omega}})$ ELSA | $\widehat{\text{CE}}_t(\hat{\boldsymbol{\omega}})$ EM-BCTS | $\widehat{\text{CE}}_t(\hat{\boldsymbol{\omega}})$ BBSL |
|---|---|---|---|---|---|---|---|---|---|
| **_CIFAR-10-LT_** | | | | | | | | | |
| ResNet-20 | $83.10_{\pm 1.15}$ | $78.24_{\pm 0.81}$ | $31.55_{\pm 1.00}$ | $35.87_{\pm 0.64}$ | $35.49_{\pm 0.58}$ | $34.90_{\pm 0.63}$ | $34.90_{\pm 0.53}$ | $35.91_{\pm 0.91}$ | $35.04_{\pm 0.78}$ |
| ResNet-32 | $85.48_{\pm 1.08}$ | $80.74_{\pm 0.77}$ | $32.77_{\pm 1.26}$ | $37.58_{\pm 0.73}$ | $39.21_{\pm 0.66}$ | $39.48_{\pm 0.70}$ | $39.35_{\pm 0.64}$ | $40.53_{\pm 0.77}$ | $39.46_{\pm 0.62}$ |
| ResNet-56 | $85.38_{\pm 1.08}$ | $81.71_{\pm 0.76}$ | $33.83_{\pm 1.24}$ | $38.01_{\pm 0.55}$ | $38.98_{\pm 0.65}$ | $38.97_{\pm 0.49}$ | $38.53_{\pm 0.64}$ | $39.38_{\pm 0.91}$ | $39.02_{\pm 0.60}$ |
| ResNet-110 | $84.94_{\pm 1.10}$ | $81.29_{\pm 0.76}$ | $34.24_{\pm 1.22}$ | $38.97_{\pm 0.63}$ | $39.66_{\pm 0.58}$ | $39.68_{\pm 0.58}$ | $39.46_{\pm 0.60}$ | $40.81_{\pm 0.81}$ | $39.68_{\pm 0.67}$ |
| **_CIFAR-100-LT_** | | | | | | | | | |
| ResNet-20 | $52.24_{\pm 1.57}$ | $44.81_{\pm 0.97}$ | $145.74_{\pm 0.54}$ | $157.00_{\pm 0.28}$ | $156.73_{\pm 0.30}$ | $147.16_{\pm 0.27}$ | $198.71_{\pm 0.35}$ | $150.78_{\pm 0.25}$ | $5083.67_{\pm 8.64}$ |
| ResNet-32 | $53.48_{\pm 1.57}$ | $47.73_{\pm 0.98}$ | $151.01_{\pm 0.60}$ | $162.27_{\pm 0.24}$ | $162.60_{\pm 0.27}$ | $154.11_{\pm 0.14}$ | $185.44_{\pm 0.25}$ | $157.72_{\pm 0.20}$ | $400.14_{\pm 0.65}$ |
| ResNet-56 | $54.21_{\pm 1.57}$ | $47.18_{\pm 0.98}$ | $151.08_{\pm 0.57}$ | $163.37_{\pm 0.19}$ | $164.01_{\pm 0.12}$ | $155.91_{\pm 0.31}$ | $509.24_{\pm 1.47}$ | $158.69_{\pm 0.23}$ | $1504.61_{\pm 1.23}$ |
| ResNet-110 | $56.58_{\pm 1.56}$ | $49.78_{\pm 0.98}$ | $153.41_{\pm 0.65}$ | $163.44_{\pm 0.13}$ | $163.65_{\pm 0.20}$ | $156.95_{\pm 0.13}$ | $169.43_{\pm 0.22}$ | $158.98_{\pm 0.13}$ | $175.28_{\pm 0.19}$ |

Table 15: Comparison of different importance weight estimators. The source is obtained with an IF $= 5$. We measure $L_1$ CE.

| Model | $\text{Acc}_s$ | $\text{Acc}_t$ | $\text{CE}_s$ | $\text{CE}_t$ | $\widehat{\text{CE}}_t(\boldsymbol{\omega}^*)$ | $\widehat{\text{CE}}_t(\hat{\boldsymbol{\omega}})$ RLLS | $\widehat{\text{CE}}_t(\hat{\boldsymbol{\omega}})$ ELSA | $\widehat{\text{CE}}_t(\hat{\boldsymbol{\omega}})$ EM-BCTS | $\widehat{\text{CE}}_t(\hat{\boldsymbol{\omega}})$ BBSL |
|---|---|---|---|---|---|---|---|---|---|
| **_CIFAR-10-LT_** | | | | | | | | | |
| ResNet-20 | $84.77_{\pm 0.99}$ | $82.77_{\pm 0.74}$ | $28.66_{\pm 0.58}$ | $29.99_{\pm 0.47}$ | $31.26_{\pm 0.55}$ | $31.75_{\pm 0.60}$ | $32.25_{\pm 0.56}$ | $32.60_{\pm 0.86}$ | $31.86_{\pm 0.62}$ |
| ResNet-32 | $86.68_{\pm 0.93}$ | $83.47_{\pm 0.73}$ | $28.84_{\pm 0.77}$ | $33.59_{\pm 0.66}$ | $32.88_{\pm 0.71}$ | $33.20_{\pm 0.67}$ | $33.03_{\pm 0.74}$ | $33.83_{\pm 0.85}$ | $33.26_{\pm 0.66}$ |
| ResNet-56 | $86.03_{\pm 0.95}$ | $84.17_{\pm 0.72}$ | $32.71_{\pm 1.45}$ | $36.76_{\pm 0.56}$ | $36.71_{\pm 0.52}$ | $35.96_{\pm 0.54}$ | $36.12_{\pm 0.53}$ | $36.67_{\pm 0.67}$ | $35.96_{\pm 0.57}$ |
| ResNet-110 | $86.44_{\pm 0.94}$ | $85.04_{\pm 0.70}$ | $33.15_{\pm 1.54}$ | $37.97_{\pm 0.85}$ | $38.98_{\pm 0.62}$ | $39.15_{\pm 0.65}$ | $39.03_{\pm 0.66}$ | $39.21_{\pm 0.74}$ | $38.92_{\pm 0.67}$ |
| **_CIFAR-100-LT_** | | | | | | | | | |
| ResNet-20 | $52.92_{\pm 1.39}$ | $50.39_{\pm 0.98}$ | $149.05_{\pm 0.58}$ | $155.00_{\pm 0.31}$ | $154.74_{\pm 0.19}$ | $153.35_{\pm 0.18}$ | $157.83_{\pm 0.16}$ | $152.64_{\pm 0.23}$ | $161.53_{\pm 0.27}$ |
| ResNet-32 | $55.52_{\pm 1.39}$ | $50.64_{\pm 0.98}$ | $155.64_{\pm 0.56}$ | $161.56_{\pm 0.27}$ | $161.79_{\pm 0.14}$ | $160.31_{\pm 0.21}$ | $163.52_{\pm 0.16}$ | $159.33_{\pm 0.16}$ | $164.22_{\pm 0.23}$ |
| ResNet-56 | $56.59_{\pm 1.38}$ | $53.33_{\pm 0.98}$ | $157.25_{\pm 0.38}$ | $163.20_{\pm 0.25}$ | $163.55_{\pm 0.18}$ | $161.88_{\pm 0.17}$ | $166.54_{\pm 0.13}$ | $160.84_{\pm 0.16}$ | $165.55_{\pm 0.21}$ |
| ResNet-110 | $57.26_{\pm 1.38}$ | $54.16_{\pm 0.98}$ | $157.92_{\pm 0.43}$ | $163.76_{\pm 0.16}$ | $164.03_{\pm 0.19}$ | $162.17_{\pm 0.16}$ | $164.27_{\pm 0.17}$ | $160.80_{\pm 0.20}$ | $165.03_{\pm 0.09}$ |

Table 16: Comparison of importance weight estimators. The source is obtained with an IF $= 2$. We measure $L_1$ CE.

| Model | $\text{Acc}_s$ | $\text{Acc}_t$ | $\text{CE}_s$ | $\text{CE}_t$ | $\widehat{\text{CE}}_t(\boldsymbol{\omega}^*)$ | $\widehat{\text{CE}}_t(\hat{\boldsymbol{\omega}})$ RLLS | $\widehat{\text{CE}}_t(\hat{\boldsymbol{\omega}})$ ELSA | $\widehat{\text{CE}}_t(\hat{\boldsymbol{\omega}})$ EM-BCTS | $\widehat{\text{CE}}_t(\hat{\boldsymbol{\omega}})$ BBSL |
|---|---|---|---|---|---|---|---|---|---|
| **_CIFAR-10-LT_** | | | | | | | | | |
| ResNet-20 | $86.74_{\pm 0.78}$ | $85.78_{\pm 0.68}$ | $22.60_{\pm 0.64}$ | $28.86_{\pm 0.40}$ | $29.90_{\pm 0.56}$ | $30.17_{\pm 0.52}$ | $30.26_{\pm 0.55}$ | $30.28_{\pm 0.50}$ | $30.26_{\pm 0.60}$ |
| ResNet-32 | $86.94_{\pm 0.78}$ | $86.82_{\pm 0.66}$ | $23.62_{\pm 0.72}$ | $32.73_{\pm 0.51}$ | $33.21_{\pm 0.56}$ | $33.04_{\pm 0.54}$ | $32.79_{\pm 0.51}$ | $32.84_{\pm 0.62}$ | $33.05_{\pm 0.55}$ |
| ResNet-56 | $88.41_{\pm 0.74}$ | $87.93_{\pm 0.64}$ | $26.15_{\pm 0.98}$ | $36.66_{\pm 0.71}$ | $36.76_{\pm 0.85}$ | $37.15_{\pm 0.93}$ | $36.92_{\pm 0.89}$ | $37.13_{\pm 0.74}$ | $37.01_{\pm 0.86}$ |
| ResNet-110 | $88.33_{\pm 0.74}$ | $87.51_{\pm 0.65}$ | $26.75_{\pm 0.87}$ | $36.80_{\pm 1.20}$ | $37.25_{\pm 1.08}$ | $38.23_{\pm 0.98}$ | $37.83_{\pm 1.01}$ | $37.37_{\pm 1.03}$ | $38.15_{\pm 1.00}$ |
| **_CIFAR-100-LT_** | | | | | | | | | |
| ResNet-20 | $56.81_{\pm 1.15}$ | $56.71_{\pm 0.97}$ | $146.17_{\pm 0.37}$ | $148.09_{\pm 0.27}$ | $147.38_{\pm 0.24}$ | $147.77_{\pm 0.23}$ | $148.02_{\pm 0.19}$ | $146.95_{\pm 0.29}$ | $181.77_{\pm 0.32}$ |
| ResNet-32 | $58.43_{\pm 1.14}$ | $58.59_{\pm 0.97}$ | $158.73_{\pm 0.29}$ | $158.92_{\pm 0.24}$ | $159.20_{\pm 0.26}$ | $159.44_{\pm 0.12}$ | $159.73_{\pm 0.20}$ | $158.42_{\pm 0.19}$ | $160.46_{\pm 0.12}$ |
| ResNet-56 | $60.42_{\pm 1.13}$ | $59.96_{\pm 0.96}$ | $161.93_{\pm 0.21}$ | $162.95_{\pm 0.15}$ | $163.12_{\pm 0.15}$ | $163.18_{\pm 0.16}$ | $163.21_{\pm 0.18}$ | $162.60_{\pm 0.14}$ | $163.19_{\pm 0.16}$ |
| ResNet-110 | $62.88_{\pm 1.12}$ | $61.99_{\pm 0.95}$ | $162.72_{\pm 0.19}$ | $163.96_{\pm 0.08}$ | $164.01_{\pm 0.17}$ | $164.06_{\pm 0.14}$ | $164.18_{\pm 0.11}$ | $163.30_{\pm 0.12}$ | $163.98_{\pm 0.16}$ |

# B  Top-label calibration

In the main paper, we focus on classwise calibration error, as it provides a more comprehensive measure of calibration by assessing the alignment of the model's confidence across all classes, rather than just the highest prediction. However, top-label calibration is widely used in the literature, so we demonstrate here how our estimator can be extended to handle this form of calibration.

For top-label calibration, we focus on the maximum score $Q = \max(f(X))$, which corresponds to the top prediction. As before, the class labels are represented as one-hot encoded variables $Y \in \{\mathbf{e}_1, \ldots, \mathbf{e}_k\} \subset \Delta^k$, where $e_i$ is the one-hot vector corresponding to class $i$. The top-label $L_p$ calibration error is [Kumar et al., 2019, Kull et al., 2019, Gruber and Buettner, 2022]:

$$\mathrm{TCE}_p(f)^p = \mathbb{E}\left[\left|\mathbb{P}\left[Y = e_{\arg\max f(X)} \mid Q\right] - Q\right|^p\right] \tag{8}$$

We aim to find an estimator of the form:

$$\widehat{\mathrm{TCE}}_p(f)^p = \frac{1}{m}\sum_{j=n+1}^{m+n}\left|\widehat{\mathbb{E}_{p_t}}[\mathbb{1}(Y = e_{\arg\max f(X)}) \mid q_j] - q_j\right|^p, \tag{9}$$

where the expectations are taken w.r.t. the target, and $q_j$ denotes the top-label prediction for input $x_j$. For the estimator of the conditional expectation we compute:

$$\mathbb{E}_{p_t}[\mathbb{1}(Y = e_{\arg\max f(X)}) \mid Q = q] \approx \frac{\frac{1}{n}\sum_{c=1}^{k}\sum_{i \in S_c}\hat{\omega}_c \kappa(Q = q, q_i)\mathbb{1}(y_i = \arg\max f(x_i))}{\frac{1}{m}\sum_{i=n+1}^{m+n}\kappa(Q = q, q_i)}, \tag{10}$$

where $S_c$ is the subset of samples where the true label is $c$, and $\mathbb{1}$ is an indicator function returning 1 if the predicted label matches the true label for sample $x_i$, and 0 otherwise.

Table 17: Amazon experiments

| Model | Uncal | TempScal | HeadToTail | EM-BCTS | CPMCN | LaSCal |
|---|---|---|---|---|---|---|
| RoBERTa | $6.03_{\pm 0.50}$ | $1.14_{\pm 0.16}$ | $0.93_{\pm 0.17}$ | $0.40_{\pm 0.11}$ | $0.37_{\pm 0.10}$ | $0.29_{\pm 0.08}$ |
| DistillRoBERTa | $10.73_{\pm 0.63}$ | $1.83_{\pm 0.25}$ | $0.58_{\pm 0.12}$ | $0.43_{\pm 0.09}$ | $0.66_{\pm 0.14}$ | $0.42_{\pm 0.11}$ |
| BERT | $17.32_{\pm 0.67}$ | $3.71_{\pm 0.35}$ | $0.73_{\pm 0.13}$ | $0.75_{\pm 0.15}$ | $2.19_{\pm 0.21}$ | $0.61_{\pm 0.12}$ |
| DistillBERT | $13.59_{\pm 0.72}$ | $2.22_{\pm 0.25}$ | $0.33_{\pm 0.10}$ | $0.37_{\pm 0.09}$ | $1.72_{\pm 0.25}$ | $0.30_{\pm 0.09}$ |
| Macro-average | $11.42_{\pm 0.59}$ | $2.23_{\pm 0.17}$ | $0.64_{\pm 0.09}$ | $0.49_{\pm 0.07}$ | $1.24_{\pm 0.19}$ | $\mathbf{0.41_{\pm 0.09}}$ |

Table 18: iWildCam experiments

| Model | Uncal | TempScal | HeadToTail | EM-BCTS | CPMCN | LaSCal |
|---|---|---|---|---|---|---|
| ResNet50 | $3.21_{\pm 0.40}$ | $1.66_{\pm 0.27}$ | $1.06_{\pm 0.24}$ | $0.64_{\pm 0.15}$ | $2.40_{\pm 0.34}$ | $0.74_{\pm 0.17}$ |
| Swin-Large | $5.92_{\pm 0.57}$ | $1.88_{\pm 0.28}$ | $1.07_{\pm 0.25}$ | $1.48_{\pm 0.27}$ | $1.17_{\pm 0.28}$ | $1.17_{\pm 0.26}$ |
| ViT-Large | $2.43_{\pm 0.41}$ | $1.59_{\pm 0.33}$ | $1.48_{\pm 0.26}$ | $3.32_{\pm 0.43}$ | $2.34_{\pm 0.33}$ | $0.62_{\pm 0.14}$ |
| ViT-Large (384) | $2.69_{\pm 0.40}$ | $1.96_{\pm 0.33}$ | $1.85_{\pm 0.34}$ | $1.93_{\pm 0.34}$ | $2.18_{\pm 0.37}$ | $0.81_{\pm 0.16}$ |
| Macro-average | $3.56_{\pm 0.41}$ | $1.77_{\pm 0.26}$ | $1.37_{\pm 0.27}$ | $1.84_{\pm 0.21}$ | $2.02_{\pm 0.23}$ | $\mathbf{0.84_{\pm 0.12}}$ |

The results presented in Tables 17 and 18 show that the observations remain the same as in the main paper: (i) LaSCal significantly reduces the CE of all models across datasets; (ii) LaSCal outperforms the baselines, achieving state-of-the-art results on the datasets and settings we experiment with.

