# OpenReview forum: "LaSCal: Label-Shift Calibration without target labels"
_NeurIPS.cc/2024/Conference — NeurIPS 2024 poster_

### Official Review · Reviewer_z7nx · 2024-06-14

**Soundness:** 3
**Presentation:** 3
**Contribution:** 3
**Rating:** 5
**Confidence:** 3

**Summary:**

The paper addresses the problem of confidence calibration under label shift given unlabeled samples from the target domain. The first step is estimating the label distribution of the target domain. Then a calibration parameter is computed using the source domain samples  where each sample is reweighted according to its label probability ratio in the source and the target domain.

**Strengths:**

The proposed method makes sense and yields good calibration results.

**Weaknesses:**

My main concern is a lack of novelty. The standard calibration methods in case of domain shift (CPCS, TransCal) are based on importance weighting of a labeled sample (x,y) from the source domain according to its similarity to the target domain p_t(x)/p_s(x). Here the proposed method applies the same principle to the label shift problem.  Given a labeled sample (x,y) from the source domain, it is reweighted according to the ration  p_t(y)/p_s(y).

**Questions:**

Can you clearly state the novelty compared to covariate shift methods?

**Limitations:**

yes

---

> ### Author Rebuttal · Authors · 2024-08-06
>
> **Can you clearly state the novelty compared to covariate shift methods?**
>
> Please see our general response for the perceived lack of novelty. To summarize:
>
> - This paper introduces the first label-free, consistent estimator for target calibration error under label shift.  Estimating calibration error under covariate and label-shift are two very different problems.
> - The re-weighting strategy in label shift is based on the label distribution ratio, which is fundamentally different from the feature distribution ratio used in covariate shift methods.
> - The paper provides both a strong theoretical justification and a thorough empirical analysis, demonstrating the effectiveness of our approach, and showcasing that existing covariate shift methods are suboptimal under label shift (see Table 2). This result emphasizes the necessity for designing recalibration methods tailored to label shift conditions.

---

> > ### Comment · Reviewer_z7nx · 2024-08-12
> >
> > I was convinced by the author explanations regarding the difference between their method and weight sampling methods for distribution shift.

---

> > > ### Author Response · Authors · 2024-08-12
> > > **Thank you**
> > >
> > > Thank you for your feedback and updated score. We’re glad that our explanation resolved your concern.

---

### Official Review · Reviewer_Lv5P · 2024-07-03

**Soundness:** 2
**Presentation:** 3
**Contribution:** 3
**Rating:** 7
**Confidence:** 4

**Summary:**

This paper proposes a novel calibration error estimator under label shifts (without ground truth). They use this estimator to apply standard calibration techniques such as temperature scaling on the unlabelled target set, allowing to calibrated the model for the target domain without needing access to a labelled set on this domain. Experiments are conducted for different strength of label shifts on various imaging and text datasets.

**Strengths:**

* [Clarity] The paper is very well-written, experiments and methods are clearly presented, figures and tables are of high quality.
* [Relevance] Estimation of calibration error under shift is an important problem, still largely under studied.
* [Experiments] Experiments are presented on multiple datasets, with statistical errors measures and contains several ablations on the main components of the proposed method. In particular ablations on the density ratio estimator shows that the method is quite robust to the particular choice of estimator.

**Weaknesses:**

**Pre-rebuttal review:**

Major:
* **Missing very important baselines for model calibration under label shift** (table 2). The authors have included multiple baselines to compare their method against, unfortunately their choice of baselines is inappropriate for the type of shift they are studying here. The problem tackled in this work is very clearly stated at the beginning of the paper, the author tackle calibration under _label shift_. However, the baselines they chose to compare against in table 2 are baselines that were designed at improving calibration under _covariate shift_ (TransCal, HeadToTail as the author state themselves in the related work section), which are two very different problems! It is unfair to compare a method specifically designed for improving calibration under label shift with methods designed to tackle covariate shift. There exists methods that are designed to recalibrate probabilities under label shift in the literature but these were completely omitted in the authors' analysis. There is no clear reason why the authors did not include these baselines, even more so since they directly use part of these works in their own method. To cite just a few missing baselines:
    * The recalibration method from Alexandari et al. In their work the authors proposes to estimate the density ratio $w= p_t(y) / p_t(y)$ via an EM algorithm and then they propose to _recalibrate the outputs_ via $\hat{p}(y=i|x) = \frac{w_i \cdot p(y=i|x)}{\sum_j w_j \cdot p(y=j|x)}$, where $w_i$ is the estimated density ratio for class i and $p(y=i|x)$ is the i-th output of the classifier for sample $x$. See section 2.2 of that paper. The authors then use these recalibrated probabilities to get updated classification predictions.
    * Similar Wen et al, in their work ‘Class Probability Matching with Calibrated Networks for Label Shift Adaption’, proposed another density ratio estimator and recalibrated the probabilities in a similar way as Alexandari et al, see section 4 of that paper. Using their re-calibrated probabilities, they achieve SOTA results in terms of accuracy under shifts.
    * **These density-ratio based recalibration baselines are the current go-to approach for re-calibration of model outputs under label shifts and should be included in the paper for a fair comparison with relevant baselines.** At least one of them needs to be included in the paper to be able to claim SOTA results in calibration under label shift.

Minor:
* **Calibration error estimator is data hungry**: in Fig 3.c. we can see that the method is very data hungry and needs at least 4000 samples to be able to perform the calibration error accurately. This should be highlighted and discussed in the limitations section, as this may hinder some practical applications.
* **Effect of recalibration step on calibration error estimation?**: I would be curious to see if the proposed calibration error estimator is able to estimate the true CE with the same precision after the proposed recalibration step compared to before. I would expect that since the error estimation is based on the estimated density ratio the estimator would be overestimating true CE after output recalibration (since recalibration would close the domain gap). Do the authors have any insights on this?


**Update after rebuttal**:

The authors have successfully addressed my concerns during the rebuttal.

**Questions:**

My main point of concern is the missing label shift recalibration baselines, please see weaknesses for details. These missing baselines are the main reason behind my rating. If the authors are able to include these baselines in Table 2 and show superiority of the proposed method over existing label shift recalibration methods, I would be willing to increase my score.

**Limitations:**

See weaknesses

---

> ### Author Rebuttal · Authors · 2024-08-01
>
> Thank you for the valuable feedback and suggestions. We implemented these baselines and performed additional experiments, which we will include in the camera-ready version. We address each question below:
>
> 1. **Add at least one more relevant baseline for model calibration under label shift**
>
> We compared our method both with CPMCN [1] and EM-BCTS [2], on CIFAR-10/100, Amazon, and iWildCam.
>
> ### CIFAR10-LT
>
> | Model| EM-BCTS| CPMCN| LaSCal |
> |-------|---------|-------|-------|
> | ResNet20  | $3.77_{\pm 0.12}$  | $3.79_{\pm 0.11}$   | $4.40_{\pm 0.15}$  |
> | ResNet32  | $4.99_{\pm 0.24}$  | $5.19_{\pm 0.21}$   | $4.78_{\pm 0.16}$  |
> | ResNet56  | $4.41_{\pm 0.11}$  | $4.42_{\pm 0.12}$   | $4.57_{\pm 0.16}$  |
> | ResNet110 | $4.40_{\pm 0.12}$  | $4.43_{\pm 0.13}$   | $4.70_{\pm 0.16}$  |
> | **Macro-average** | **4.39$_{\pm 0.22}$** | $4.46_{\pm 0.20}$ | $4.61_{\pm 0.16}$ |
>
> ### CIFAR-100-LT
>
> | Model | EM-BCTS | CPMCN| LaSCal|
> |--------|------|------|----------|
> | ResNet20  | $25.01_{\pm 0.21}$  | $25.02_{\pm 0.24}$   | $5.61_{\pm 0.08}$  |
> | ResNet32  | $26.17_{\pm 0.21}$  | $24.76_{\pm 0.20}$   | $5.79_{\pm 0.08}$  |
> | ResNet56  | $26.33_{\pm 0.23}$  | $24.53_{\pm 0.22}$   | $5.89_{\pm 0.07}$  |
> | ResNet110 | $28.22_{\pm 0.24}$  | $26.49_{\pm 0.22}$   | $6.19_{\pm 0.07}$  |
> | **Macro-average** | $26.43_{\pm 0.25}$ | $25.20_{\pm 0.22}$ | **5.87$_{\pm 0.08}$** |
>
> ### Amazon
>
> | Model  | EM-BCTS | CPMCN    | LaSCal|
> |----|----|----|----|
> | RoBERTa          | $2.72_{\pm 0.35}$  | $1.36_{\pm 0.17}$   | $3.64_{\pm 0.32}$  |
> | DistillRoBERTa   | $2.13_{\pm 0.28}$  | $2.81_{\pm 0.23}$   | $2.71_{\pm 0.25}$  |
> | BERT             | $3.95_{\pm 0.40}$  | $9.32_{\pm 0.54}$   | $3.74_{\pm 0.39}$  |
> | DistillBERT      | $3.41_{\pm 0.36}$  | $5.48_{\pm 0.34}$   | $3.40_{\pm 0.28}$  |
> | **Macro-average**| **3.05$_{\pm 0.36}$** | $4.74_{\pm 0.42}$ | $3.37_{\pm 0.31}$|
>
> ### iWildCam
>
> | Model | EM-BCTS| CPMCN | LaSCal |
> |----|---|---|----|
> | ResNet50         | $15.84_{\pm 0.57}$  | $19.43_{\pm 0.69}$   | $13.01_{\pm 0.45}$ |
> | Swin-Large       | $16.81_{\pm 0.63}$  | $18.03_{\pm 0.62}$   | $15.19_{\pm 0.48}$ |
> | ViT-Large        | $24.83_{\pm 1.31}$  | $19.33_{\pm 0.80}$   | $13.00_{\pm 0.43}$ |
> | ViT-Large (384)  | $19.78_{\pm 0.73}$  | $20.74_{\pm 0.72}$   | $16.58_{\pm 0.69}$ |
> | **Macro-average**| $19.31_{\pm 0.76}$      | $19.38_{\pm 0.75}$         | **14.45$_{\pm 0.52}$**          |
>
> Overall, on all datasets LaSCal either outperforms or performs competitively with both CPMCN [1] and EM-BCTS [2], as per the macro-average CE across models (note for CIFAR-10 and Amazon the error bars overlap for LaSCal and EM-BCTS). LaSCal's gains are prominent for datasets which feature a large(r) number of classes (100 in CIFAR-100 and 20 in iWildCam). We hypothesize this is due to the increased complexity of the optimization process associated with higher-dimensional spaces, which our approach seems to handle more effectively. Aside from the empirical gains, our method has the following advantages compared to these methods: (1) is based on a consistent estimator of target calibration error; (2) it enables unsupervised calibration on the target distribution, whereas CPMCN [1] and EM-BCTS [2] perform the calibration step on a labeled validation set (from source).
>
> 2. **Calibration error estimator is perceived as data-hungry**
>
> While it is true that the estimator requires a sufficient number of data samples (4000 samples in Fig. 3c) to accurately estimate the calibration error (CE), we do not believe that this should be perceived as data-hungry. We do agree that in severely data-scarce settings, this requirement may limit potential applications, and we will discuss this in the revised manuscript.
> However, please note that the error rate of our estimator is $O(n^{-1/2} + m^{-1/2})$, which is the same as the weight estimation methods (see Lemma1 for the RLLS estimator in *Azizzadenesheli et al.* [6], and top paragraph on page 8 in *Garg et al.* [4] for the EM-BCTS method from *Alexandari* et al. [2]). Therefore, the data requirement is not unique to our method, but rather is common across all weight estimation-based approaches. Further, Fig. 3c also shows that our estimator has a positive bias in scenarios with limited data. This characteristic is preferable as it prevents the false impression that a model is well-calibrated due to insufficient sample size. In essence, our method errs on the side of caution, ensuring reliability even in data-constrained environments.
>
> 3. **Effect of recalibration step on calibration error estimation?**
>
> Thank you for the interesting question. The label shift gap is not affected by the recalibration step, and therefore the obtained weights from the weight estimation methods remain the same before and after calibration. Having said that, we hypothesize that the precision of our estimator would remain the same. To gain insights, we performed additional experiments on CIFAR-10, comparing the precision of estimating CE using our estimator, compared to ground truth (with labels) before and after the re-calibration step with LaSCal. The empirical results we obtained seem to confirm our hypothesis.
>
> | Model | Uncal | LaSCal |
> |-----|-----|---|
> | ResNet20 (w/ labels) | $9.01_{\pm 0.34}$  | $4.43_{\pm 0.16}$  |
> | ResNet20 (w/o labels) | $9.15_{\pm 0.50}$ | $4.48_{\pm 0.25}$  |
> | **Absolute Difference** | **0.14**| **0.05**|
> | ResNet32 (w/ labels)  | $10.41_{\pm 0.44}$ | $4.76_{\pm 0.13}$  |
> | ResNet32 (w/o labels) | $11.94_{\pm 0.49}$ | $6.05_{\pm 0.41}$  |
> | **Absolute Difference** | **1.53**  | **1.29** |
> | ResNet56 (w/ labels) | $11.18_{\pm 0.23}$ | $4.56_{\pm 0.14}$  |
> | ResNet56 (w/o labels)| $11.63_{\pm 0.32}$ | $4.99_{\pm 0.17}$  |
> | **Absolute Difference** | **0.45**| **0.43** |
> | ResNet110 (w/ labels)   | $11.86_{\pm 0.26}$ | $4.71_{\pm 0.14}$ |
> | ResNet110 (w/o labels)  | $12.15_{\pm 0.29}$ | $4.98_{\pm 0.18}$ |
> | **Absolute Difference** | **0.29**| **0.27** |

---

> > ### Comment · Reviewer_Lv5P · 2024-08-08
> >
> > I thank the authors for their extensive response and additional experimental results. I have updated my score accordingly.
> > My primary concern i.e.  the need to compare with relevant label adaptation baselines has been addressed by the provided experiments and substantially strengthen the paper. Effect of recalibration step on calibration error estimation: great to see these additional results and that the ECE estimation still holds. I agree with the rationale of the authors for point 2. and hope that this discussion will be added to the manuscript.
> > I have no further question at this point.

---

> > > ### Author Response · Authors · 2024-08-09
> > > **Thank you**
> > >
> > > Thank you for your valuable feedback, as well as voting for acceptance of our paper. We are glad our response has addressed your concerns, and we firmly believe that your suggestions greatly improved the manuscript. We will update the manuscript based on the new results and discussion we had during the rebuttal. Thank you again!

---

### Official Review · Reviewer_Mi4F · 2024-07-05

**Soundness:** 3
**Presentation:** 3
**Contribution:** 3
**Rating:** 6
**Confidence:** 4

**Summary:**

This work considers the problem of model calibration under label shift with label-free data. The work obtains the unsupervised calibration error through the kernel-based estimation of the TARGET data with the SOURCE data and applies it to the TS calibration method. The problem's has some novelty and is technically feasible.

**Strengths:**

1. The article considers the problem of model calibration in unsupervised and label shift for unlabeled data. The problem seems to be interesting and has some novelty.
2. The article provides a solution to the problem of model calibration under unlabeled data by estimating the calibration error for unlabeled data through a kernel method, which is meaningful.

**Weaknesses:**

The other methods in the comparison experiments, such as TS, use the source data. Such comparison experiment seems to be not rigorous. In comparison experiments, the validation sets for different comparison methods should remain the same.

**Questions:**

1. What are the potential causes of label shift?
2. Does label shift have an impact on model calibration? Label shift between training and validation sets does not seem to be reflected by calibration. Can the authors give some explanation for the relationship between label drift and model calibration? In my opinion, calibration methods for unsupervised data are the focus of this work.
3. Does the result obtained from calibrating the model with a validation set that has data shift negatively affect the results of the training set?
4. See weaknesses.

**Limitations:**

see Weaknesses and Questions.

---

> ### Author Rebuttal · Authors · 2024-08-06
>
> We thank the reviewer for the positive review and feedback on our paper. We provide an explanation to the questions below:
>
> 1. **Some of the baselines use source data for calibration.**
>
> While it is true that compared to i.i.d calibration methods, unsupervised calibration methods (LaSCal, HeadToTail, TransCal, CPCS) have access to samples from the unlabeled target distribution in addition to the labeled validation set (from source), the reasons we included these methods are two-fold:
>
> 1. To showcase that traditional i.i.d. calibration methods fall short in the case of label shift and highlight the necessity for designing calibration methods specifically for this scenario.
>
> 2. To be consistent with the papers on calibration under covariate shift i.e., HeadToTail, TransCal and CPCS, which also include TS, Vector Scaling, Ensemble TS and IROvA.
>
> In the revised manuscript, we propose to replace two of these baselines (Vector scaling and ETS) with the baselines proposed by Reviewer **Lv5P** in Table 2, and move the VectScal and ETS to Appendix. We will also group the methods or add markers, to make this difference more visible in the tables.
>
> 2. **What are the potential causes of label shift?**
>
> Some of the potential causes of label shift:
>
> - Change in population demographics: e.g. a model trained on mainly adult population (Hospital A) is deployed in a hospital that primarily serves children (Hospital B).
> - Unusual events like disease outbreaks or natural disasters: e.g. during a pneumonia outbreak, $p(Y)$ (e.g. flu) might rise, but the symptoms of the disease $p(X| Y)$ (e.g. cough given flu) do not change.
> - Geographic changes: e.g. consider wildlife classification model for identifying animal species based on camera trap images (as in our experiments), trained on images from one region with a particular animal species distribution, and deployed in a different region with a different animal speciries distribution.
> - Seasonal changes: A model designed to predict seasonal allergies might face label shift if it is trained using data collected during spring, when pollen levels are high and hence allergies are more frequent, and then applied in autumn when allergies are less frequent
>
> 3. **Does label shift have an impact on model calibration? Label shift between training and validation sets does not seem to be reflected by calibration.**
>
> Model calibration is defined with respect to a data distribution. When that data distribution changes (e.g., when facing label shift), the predicted probabilities of a model trained on the source distribution may no longer reflect the true empirical frequencies of the target distribution, leading to poor calibration. For instance, if a model trained on a skewed dataset predicts a high probability for a rare class, this probability might not accurately reflect the true frequency of that class (conditioned on that probability) in the new distribution where that class is more frequent.
>
> Tables 10-16 in the Appendix confirm that label shift does have an impact on model calibration. For example, in Table 16 we can observe a notable increase of the calibration error from source data (CE_s) to the label-shifted target data (CE_t), particularly prominent for the Amazon dataset. Please note that in our experiments, the training and validation sets (CE_s) are sampled from the source distribution, while the test set (CE_t) is sampled from the label-shifted target distribution.
>
> 4. **Does the result obtained from calibrating the model with a validation set that has data shift negatively affect the results of the training set?**
>
> Our proposed post-hoc calibration method is accuracy-preserving, as it performs temperature scaling by optimizing the CE estimator. Since the logits are scaled with a single parameter, temperature scaling maintains the original order of the predictions, ensuring that the predicted class remains unchanged. Therefore, calibration on label-shifted data does not negatively affect the predictive performance of the classifier on the training (source) set.

---

> > ### Comment · Reviewer_Mi4F · 2024-08-08
> > **Response to Rebuttal**
> >
> > I appreciate the author's response. I do not have any additional questions at this time.

---

> > > ### Author Response · Authors · 2024-08-09
> > > **Thank you**
> > >
> > > Thank you for your time and valuable feedback. We are glad that our responses addressed your questions and we will update the manuscript with extra discussion where appropriate.

---

### Official Review · Reviewer_vmtH · 2024-07-09

**Soundness:** 4
**Presentation:** 3
**Contribution:** 2
**Rating:** 6
**Confidence:** 4

**Summary:**

This paper proposes a consistent estimator of class-wise expected calibration error (class-wise ECE) for unsupervised domain adaptation under label shift assumption, i.e., the class proportion of the source $p_s(y)$ and target distribution $p_t(y)$ differs while the class-conditional probability $p(X|y)$ remains the same. In this problem, estimating class-wise ECE is not straightforward as target domain data is unlabeled. The proposed method LaSCal suggests estimating importance weight using existing methods and then incorporate it in the validation objective. Once we have validation objective, one can use a simple post-hoc method such as temperature scaling to tune the temperature  to minimize validation class-wise ECE objective. Experiments show that the proposed consistent estimator performs better than baselines that were not designed for label shift scenario.

**Strengths:**

1. The proposed method has strong theoretical justification because it is a consistent estimator of the target classwise-ECE.
2. Experiments clearly showed that the proposed method is effective.
3. Not only does the proposed method show the best performance in experiments, but several different configurations of the proposed methods were also analyzed (e.g., using different importance weight methods, sensitivity analysis of different ratio of positive/negative, source/target, sample size).

**Weaknesses:**

1. Given existing techniques for learning under label shift, I might be wrong to find the proposed method is quite straightforward without much difficulty because the estimator is based on the well-known importance weighting method. Having said that, conducting experiments and analyses is still very important and useful as label shift learning is one important kind of dataset shift in unsupervised domain adaptation.
2. Only classwise-ECE metric is considered for the objective function, although there are many different metrics for calibration error.

**Questions:**

1. Is it possible to extend the proposed techniques for tuning different objectives rather than class-wise ECE, e.g., expected calibration error (ECE). I feel that it might also be useful to look into this setting or discuss it in future work. Intuitively, I think it should be possible without much modification from the proposed method. Given the proposed method relies on the accuracy of importance weights, analyzing different objectives under these constraints can also give some insights for reliable confidence learning in this setting.
2. In (7), it seems we only use target data for the objective. We only used source data for importance weight estimation. I was wondering if it is also possible to consider incorporate source data in this objective as well to improve the performance. What do you think?

**Limitations:**

The discussions of limitations and future work are appropriate.

---

> ### Author Rebuttal · Authors · 2024-08-01
>
> Thank you for the valuable feedback and future work suggestions. We performed further experiments which serve as an interesting addition to our paper. Regarding the questions/concerns:
>
> 1. **Proposed method is perceived as straightforward**: While our method builds upon the well-known importance weighting technique, our contribution extends beyond the straightforward application of this method.
>
> - We derive the first consistent calibration error estimator under label shift, without using target labels. Furthermore, the estimator has a known error rate $O(n^{-1/2} + m^{-1/2})$ [L140]. Depending on the choice of kernel, the estimator can be differentiable and integrated as an objective in both post-hoc and trainable calibration methods. To the best of our knowledge, no other estimator exists for target calibration error under label shift with the same properties as ours.
>
> - Based on this estimator, we introduce an accuracy-preserving recalibration method. While some of the papers addressing label shift (*CPMCN, Wen et al. [1]; EM-BCTS, Alexandari et al. [2]*), do incorporate a calibration step, it is performed on validation (source) data, whereas our method performs unsupervised calibration on the target domain.
>
> - We provide a thorough analysis of our method’s properties, demonstrating its robustness across datasets, modalities and severity of shift. We benchmark different weight estimation methods and identify the most robust method for our task. Note that although several weight estimation techniques exist for the label shift scenario, none of them extend their application beyond improving the predictive performance of the classifier.
>
> 2. **Alternative objective functions and/or metrics:** Indeed, our method (LaSCal) naturally extends to the standard ECE. In an additional experiment, we adapted our label-free estimator under label shift to estimate ECE and used that as an objective for temperature scaling. The table below reports ECE (with labels) for several baselines: IID temperature scaling, HeadToTail (as representative for methods derived under covariate shift assumption) and the two new baselines proposed by reviewer **Lv5P**, which include most recent works (*CPMCN [1] @ ICLR 2024*) on calibrating models facing label shift. Please see our general response for reasons why we focus on classwise ECE.
>
> ### Amazon experiments
>
> | Model  | Uncal | TempScale  | HeadToTail      | EM-BCTS      | CPMCN | LaSCal             |
> |------|--------|-----------|------------|----------|-----------|---------|
> | RoBERTa            | $6.03_{\pm 0.50}$  | $1.14_{\pm 0.16}$  | $0.93_{\pm 0.17}$  | $0.40_{\pm 0.11}$  | $0.37_{\pm 0.10}$    | $0.29_{\pm 0.08}$  |
> | DistillRoBERTa | $10.73_{\pm 0.63}$ | $1.83_{\pm 0.25}$  | $0.58_{\pm 0.12}$  | $0.43_{\pm 0.09}$  | $0.66_{\pm 0.14}$    | $0.42_{\pm 0.11}$  |
> | BERT            | $17.32_{\pm 0.67}$ | $3.71_{\pm 0.35}$  | $0.73_{\pm 0.13}$  | $0.75_{\pm 0.15}$  | $2.19_{\pm 0.21}$    | $0.61_{\pm 0.12}$  |
> | DistillBERT    | $13.59_{\pm 0.72}$ | $2.22_{\pm 0.25}$  | $0.33_{\pm 0.10}$  | $0.37_{\pm 0.09}$  | $1.72_{\pm 0.25}$    | $0.30_{\pm 0.09}$  |
> | **Macro-average**  | $11.42_{\pm 0.59}$ | $2.23_{\pm 0.17}$| $0.64_{\pm 0.09}$  | $0.49_{\pm 0.07}$ | $1.24_{\pm 0.19}$ | **0.41$_{\pm 0.09}$** |
>
> ### iWildCam experiments
>
> | Model | Uncal | TempScale| HeadToTail | EM-BCTS | CPMCN | LaSCal |
> |-------|---------|-------|-------------|-----------|---------------|---------|
> | ResNet50         | $3.21_{\pm 0.40}$  | $1.66_{\pm 0.27}$  | $1.06_{\pm 0.24}$  | $0.64_{\pm 0.15}$  | $2.40_{\pm 0.34}$    | $0.74_{\pm 0.17}$  |
> | Swin-Large       | $5.92_{\pm 0.57}$  | $1.88_{\pm 0.28}$  | $1.07_{\pm 0.25}$  | $1.48_{\pm 0.27}$  | $1.17_{\pm 0.28}$    | $1.17_{\pm 0.26}$  |
> | ViT-Large        | $2.43_{\pm 0.41}$  | $1.59_{\pm 0.33}$  | $1.48_{\pm 0.26}$  | $3.32_{\pm 0.43}$  | $2.34_{\pm 0.33}$    | $0.62_{\pm 0.14}$  |
> | Vit-Large (384)    | $2.69_{\pm 0.40}$  | $1.96_{\pm 0.33}$  | $1.85_{\pm 0.34}$  | $1.93_{\pm 0.34}$  | $2.18_{\pm 0.37}$    | $0.81_{\pm 0.16}$  |
> | **Macro-average** | $3.56_{\pm 0.41}$ | $1.77_{\pm 0.26}$ | $1.37_{\pm 0.27}$  | $1.84_{\pm 0.21}$ | $2.02_{\pm 0.23}$ | **0.84$_{\pm 0.12}$** |
>
> From the results we see that the observations remain the same as in the main paper: (1) LaSCal significantly reduces the CE of all models across datasets; (2) LaSCal outperforms the baselines, achieving state-of-the-art results on the datasets and settings we experiment with. We will add a summary of these results in the camera-ready version of the paper (and the full experiments in Appendix).
>
> 3. **Adding source data to the objective**
>
> Incorporating source data could be useful in the scenario where $p_s(X) = p_t(X)$ (please note that this is NOT implied by the label shift definition). We tested this approach empirically by measuring CE on the target distribution of CIFAR-10 using 3 different estimators: (1) Ground truth (with labels); (2) Our estimator (no labels, no source samples); (3) Our estimator + source samples (no labels, with source samples). The results (reported in the Table below) demonstrate that this approach leads to less precise estimates, i.e., yields higher estimated values compared to ground truth (computed with labels).
>
> | Model      | Ground truth      | Our estimator     | Our estimator + source samples  |
> |---------|----------------|--------------------|--------------------|
> | ResNet-20  | $8.95_{\pm 0.36}$  | $9.08_{\pm 0.53}$  | $11.55_{\pm 0.52}$ |
> | ResNet-32  | $10.46_{\pm 0.25}$ | $11.97_{\pm 0.71}$ | $15.37_{\pm 0.66}$ |
> | ResNet-56  | $11.18_{\pm 0.29}$ | $11.73_{\pm 0.48}$ | $15.06_{\pm 0.50}$ |
> | ResNet-110 | $11.92_{\pm 0.40}$ | $12.20_{\pm 0.39}$ | $15.27_{\pm 0.59}$ |
>
> We also incorporated the estimator including source samples as an objective function for post-hoc calibration (with the temperature scaling method), and we observed no improvement in the results compared to those reported in the paper.

---

> > ### Comment · Reviewer_vmtH · 2024-08-12
> > **Thank you very much for the paper update and conducting requested experiments**
> >
> > I have read other reviews and rebuttals. I appreciate the authors responding to my concerns and conducting more experiments on using source data. I raised my score to 6 (Weak accept) mainly because of the experiments added and the ECE results.

---

> > > ### Author Response · Authors · 2024-08-12
> > > **Thank you**
> > >
> > > Thank you for revisiting our paper and for your updated evaluation. We're glad that the additional experiments addressed your concerns.

---

### Author Rebuttal · Authors · 2024-08-06

We want to thank the reviewers for the constructive feedback on our paper. We appreciate the provided insights and are pleased to see the recognition of several strengths in our work. Below, we summarize the main strengths and weaknesses highlighted by the reviewers, and address them accordingly.

### Strenghts

- **Theoretical justification and novelty:** Our method has been acknowledged for its strong theoretical foundation as a consistent estimator of the target classwise-ECE (**vmtH**). Additionally, the novelty of addressing unsupervised model calibration under label shift was noted as a meaningful and interesting problem (**Mi4F**).
- **Comprehensive experiments**: The comprehensiveness and clarity of our experiments are well-received by most reviewers. Reviewer **vmtH** noted that our experiments clearly showed the effectiveness and superior performance of our method, while reviewer **Lv5P** praised the robustness of our method. Additionally, both reviewers (**vmtH** and **Lv5P**) compliment our clear presentation and analysis which includes multiple datasets, statistical error measures, and thorough ablations.
- **Clarity and presentation:** The clarity and quality of writing, figures, and tables were praised (**Lv5P**), highlighting that our paper effectively communicates the methodology and findings.

### Weaknesses and response

- **Perceived lack of novelty**: Reviewer **z7nx** has concerns that our method, based on importance weighting, could be lacking novelty compared to covariate shift methods.


We would like to emphasize that our work introduces the first label-free, **consistent** estimator of target calibration error under **label shift**. Estimating calibration error under covariate and label shift are two very different problems (also pointed out by reviewer **Lv5P**). Our estimator has a known error rate, and is differentiable when used with a differentiable kernel (*Popordanoska et al. [5]*). **To the best of our knowledge, no other estimator exists for target calibration error under label shift with the same properties as ours**. Furthermore, while importance weighting is a common technique for addressing distribution shifts, the re-weighting in label shift is based on the label distribution ratio $p_t(y) / p_s(y)$, which is fundamentally different from the feature distribution ratio $p_t(x)/p_s(x)$ used in covariate shift methods. Our paper provides both a strong theoretical justification (reviewer **vmtH**) and a comprehensive empirical analysis demonstrating the robustness and clear advantage over covariate shift methods in these conditions. Specifically, Table 2 in the main paper shows that existing methods derived under covariate shift assumption (HeadToTail, CPCS, TransCal) are suboptimal under label shift, which further highlights the need for designing CE estimators and recalibration methods tailored to this type of shift.

- **New baselines**: Reviewer **Lv5P** suggested including two relevant baselines designed for label shift scenarios.

We conducted experiments with the proposed baselines (CPMCN [1], EM-BCTS [2]) and observed that for all dataset / model combinations, LasCal either significantly outperforms or remains competitive with them; therefore these new experiments do not affect the overall conclusions drawn from our paper. Please note that these baselines perform a calibration step on a labeled validation (source) data prior to obtaining the importance weights, whereas our approach allows for unsupervised calibration on the target distribution. See the response to reviewer **Lv5P** for details.

- **Choice of metric**: Reviewer **vmtH** asked if we can extend the proposed technique to different objectives rather than classwise- ECE, e.g. expected calibration error ECE.

We focus on classwise calibration error because it is a stronger notion of calibration compared to ECE (i.e., CWCE = 0 implies ECE = 0, but not the other way around; see Theorem 3.1 in Gruber and Buettner [3]). Note that, we already report ECE in the reliability diagrams in Figure 2 (and Fig. 6 in Appendix), which show that optimizing our CWCE estimator leads to superior performance also in terms of ECE compared to competing methods. As per the reviewer’s suggestion, we adapted our estimator to estimate target ECE without target labels, and performed experiments for the Wilds datasets (iWildCam and Amazon) using ECE as a calibration objective. Overall, our insights and conclusions remain the same as with the original submission. Please see the response to **vmtH** for details and experiments.


*[1] Wen et. al., 2024 @ ICRL: Class Probability Matching with Calibrated Networks for Label Shift Adaption*

*[2] Alexandari et. al., 2020 @ ICML: Maximum Likelihood with Bias-Corrected Calibration is Hard-To-Beat at Label Shift Adaptation*

*[3] Gruber and Buettner 2022 @ NeurIPS: Better Uncertainty Calibration via Proper Scores for Classification and Beyond*

*[4] Garg et al., 2020 @ NeurIPS:  A Unified View of Label Shift Estimation*

*[5] Popordanoska et al., 2022 @ NeurIPS: A consistent and differentiable Lp canonical calibration error estimator*

*[6] Azizzadenesheli et al., 2019 @ ICLR: Regularized Learning for Domain Adaptation under Label Shifts*

---

### Decision · Program_Chairs · 2024-09-25

**Decision:**

Accept (poster)

**Comment:**

This paper introduces a novel calibration error estimator designed for unsupervised domain adaptation under label shift, where class proportions differ between source and target domains, but class-conditional probabilities remain consistent. The proposed method, LaSCal, estimates class-wise Expected Calibration Error (ECE) without needing labeled data from the target domain. LaSCal achieves this by estimating importance weights and incorporating them into a validation objective, which is then minimized using post-hoc techniques like temperature scaling. Experiments demonstrate that LaSCal outperforms existing baselines not tailored for label shift scenarios.

The proposed method is supported by strong theoretical foundations and its effectiveness is thoroughly validated through empirical experiments. Additionally, the authors have effectively addressed most of the concerns raised by the reviewers, leading to a positive consensus. Therefore, I also recommend accepting this paper.